# Long spin lifetimes of charge carriers in rubrene crystals due to fast transient-localization motion

Remington L. Carey[1,3], Xinglong Ren [1,3] ✉, Ian E. Jacobs [1], Jan Elsner [2], Sam Schott[1], Elliot Goldberg [1], Zichen Wang [1], Jochen Blumberger [2] & Henning Sirringhaus [1] ✉

Field-induced electron spin resonance provides valuable insights into the interplay between spin and charge dynamics in organic semiconductors. We apply this technique to ion-gel-gated capacitors and conventional field-effect transistors to study the temperature-dependent carrier dynamics of high-mobility rubrene single-crystals. Unlike previous measurements on other molecular and polymer semiconductors, we observe remarkably long spin relaxation times–on the order of microseconds–persisting from room temperature down to 15 K. Such long relaxation times are caused by the rapid transient-localization motion of charge carriers, which induces efficient motional narrowing. Additionally, by leveraging the high injection efficiency of ion-gel-gated devices, we observe spin lifetimes shortening at high carrier concentrations. This is attributed to emerging spin-spin dipolar interactions and can be modelled using an approach adapted from fluid-phase nuclear magnetic resonance. Our work demonstrates that field-induced electron spin resonance provides a powerful probe of the transient-localization physics of high-mobility molecular crystals.

In organic semiconductor crystals, charge transport can be described via a model known as transient localization, which focuses on the effects of dynamic disorder induced by thermal lattice vibrations. In the model, charge carriers transiently localize in their dynamically disordered electronic landscape on timescales typically shorter than 100 femtoseconds, but undergo diffusive motion on longer timescales as the molecular lattice evolves[1]. The wavefunction of a charge carrier is considered to be highly dynamic, switching between highly localized and highly delocalized configurations on timescales ranging from 10 to 100 femtoseconds. By coupling to vibrational modes, the wavefunction occasionally reaches configurations where it is delocalized over 10–100 molecules, then re-localizes to a different region; it is this process that governs long-range charge diffusion and carrier mobility[2]. Transient localization is considered to be the most appropriate

theoretical framework for understanding charge dynamics in high-mobility molecular crystals in the intermediate regime between localized hopping and extended-state band transport, and explains many of the experimentally observed charge-transport phenomena[3]. Relevant here is that the theory has important implications not just for charge dynamics, but also for the associated spin dynamics. These have not been explored yet in much depth, but they could provide further fundamental insight into this unique charge-transport regime.

Measurements of spin dynamics often use the longitudinal and transverse relaxation times, $T_1$ and $T_2$, to establish links between charge and spin transport. $T_1$, also called the spin-lattice relaxation time, characterizes how quickly (or slowly) a paramagnetic material loses its magnetization along the quantization axis when placed in and subsequently removed from an external magnetic field Though details

[1]Cavendish Laboratory, University of Cambridge, Cambridge, United Kingdom. [2]Department of Physics and Astronomy and Thomas Young Centre, University College London, London, United Kingdom. [3]These authors contributed equally: Remington L. Carey, Xinglong Ren. ✉e-mail: xr216@cam.ac.uk; hs220@cam.ac.uk

vary based on molecular structure, this loss of magnetization is typically the result of energy-exchanging spin flips between electrons and the surrounding lattice. $T_2$, on the other hand, is known as the spin-spin relaxation time and characterizes the decay of the transverse magnetization that precesses around the quantization axis. Though many processes can contribute to it in a given system, it is most generally due to spins losing phase coherence between one another. A third relaxation time, $T_2^*$, is also often used to characterize a system. This term captures the contribution to $T_2$ that is not a result of spin flips ($T_1$), and is known as the pure dephasing or decoherence term; it is typically driven by different spins being located in different magnetic environments and thus precessing at different rates.

Field-induced electron spin resonance (FI-ESR) is a powerful probe of spin relaxation that has been used extensively to establish intricate links between charge and spin physics in organic field-effect transistors (OFETs)[4–15]. Compared with other techniques, such as spin-valves or spin-pumping measurements, which may suffer from device-related artifacts[16–21], ESR provides a more direct way to determine spin lifetimes by analyzing the dependence of the ESR signal on microwave power and magnetic field (see Supplementary Note 1 for a discussion of different techniques)[22]. Of note here are measurements on the thiophene-based molecular crystal C10-DNBDT-NW, for which spin relaxation has been attributed to an Elliot-Yafet momentum-scattering process similar to the mechanism that occurs in band transport for inorganic semiconductors. Even though charge-carrier mobility remains high in these systems at room temperature—exceeding $10 \, cm^2 \, V^{-1}s^{-1}$—this momentum-scattering process still limits spin lifetimes to shorter than 100 nanoseconds[15]. Interestingly, very similar room-temperature spin-relaxation behavior has also been observed in conjugated polymers[9], although the relaxation mechanism in this case has been attributed to spatial rather than momentum scattering. It therefore appears that, although the (room-temperature) relaxation pathways in molecular crystalline versus polymeric semiconductors operate via different mechanisms, the two scattering processes result in a similar decrease in relaxation times with increasing temperature. Equally interesting is that, at temperatures typically between 100 and 200 K, the same motional narrowing relaxation process has been observed in both molecular and polymeric systems[4,5,7,9,23–25]. While (spatial or momentum) scattering shortens both $T_1$ and $T_2$ with increasing temperature, motional narrowing lengthens $T_2$ because fast-moving charges are able to average out local variations in magnetic-field environments. Considering these different relaxation mechanisms, the question naturally arises of what type of charge transport is most conducive to long spin lifetimes, i.e., how can scattering be reduced and motional narrowing enhanced to achieve long relaxation times even at room temperature?

The high-temperature scattering process limiting relaxation times in C10-DNBDT-NW is caused by the spin-orbit interaction, which is the coupling a spin feels to the relativistic magnetic field induced by the charge's motion in the electric field of a nearby nucleon. Spin-orbit effects are notoriously weak in hydrocarbon-based molecules, meaning these molecules should also have correspondingly long spin lifetimes. However, experimental observation of long spin lifetimes at room temperature is exceptionally rare. Graphene, one of the most well-known carbon materials, is theoretically predicted to exhibit spin lifetimes of microseconds to milliseconds due to its weak intrinsic spin-orbit coupling and negligible hyperfine interaction, and it is indeed one of the most extensively studied materials in spintronics[26,27]. Yet, experimentally measured spin lifetimes in graphene are orders of magnitude smaller than those predictions, with maximum values on the order of 10 ns[28]. This gap between theory and experiment has been attributed to both extrinsic factors (e.g., substrate-induced electron-hole puddles[29]) and unusual spin-relaxation mechanisms (e.g., resonant scattering due to magnetic impurities[30], spin-pseudospin entanglement[31]). These explanations are based on the unique

properties of graphene, implying that the difficulty in achieving long spin lifetimes in graphene does not rule out the possibility of doing so in other organic semiconductors. Rubrene holds the distinction of having the highest mobility among the hydrocarbons, making it a natural candidate for such studies[32]: its weak spin-orbit interaction should prevent relaxation times from being shortened by scattering processes, and its high mobility should allow motional narrowing to increase relaxation times. Unfortunately, testing this hypothesis via temperature-dependent FI-ESR on rubrene single crystals is experimentally challenging: large crystals are required to inject the minimum number of spins detectable in ESR, yet such large crystals tend to break under thermal strain.

Here, we report results overcoming this issue by using ion gels as gate dielectrics in lieu of traditional silicon oxide or polymer dielectrics. Doing so allowed us to achieve capacitances of $1–100 \, \mu F \, cm^{-2}$[33,34] over the full temperature range, which corresponds to the injection of up to $10^{13}$ charges $cm^{-2}$ at ~1 V—well above the ESR detection threshold for the typical crystal size of $0.1 \, cm^2$. To compare and extend our results to more conventional rubrene-based FETs, we also report measurements on devices gated with conventional silicon dioxide and Cytop, which required integration of multiple thin crystals into a device to achieve sufficient sensitivity. In both device architectures, we consistently observed that transient-localization motion gives rise to highly effective motional narrowing and very long spin lifetimes. Moreover, the ion-gel-gated devices also allowed us to probe the onset of spin-spin interactions in rubrene at high carrier densities.

## Results

### Effective motional narrowing in ion-gated rubrene

For our ion-gel-gated devices, we used a two-contact, side-gated capacitor architecture with an ion gel based on 1-butyl-1-methylpyrrolidinium bis(trifluoromethylsulfonyl)imide ([BMP][TFSI]) and the polymer poly(vinylidenefluoride-co-hexafluoropropylene) (PVDF-HFP) (see Methods); Fig. 1a shows a schematic (left) and photograph (right) of the sample. Initial FI-ESR scans showed that the FI-ESR signal increases with gate voltage due to increasing carrier concentration, as expected (Fig. 1b). When measured at $V_g = -1.2 \, V$ and 290 K (Fig. 1c), we extracted a g-factor of 2.0024, which confirmed injection of holes into the rubrene channel[11,23,35,36]. We note that the extracted derivative peak-to-peak linewidth of $B_{pp} = 0.05 \, G$ is exceptionally narrow, indicating a very long spin lifetime. Despite the intensity of the ESR signal growing with gate voltage, we did not record measurements taken with $V_g < -1.5 \, V$ due to the onset of an irreversible electrochemical reaction that produced an additional resonance line. (See Supplementary Note 2).

Transverse ($T_2$) and longitudinal ($T_1$) relaxation times were extracted by measuring spectra as a function of applied microwave power. $T_2$ manifests as a broadening of the resonance signal due to variations in local fields across the spin ensemble, while $T_1$ is measured by increasing microwave power to the point that spins cannot relax quickly enough to balance excitation by the field, which results in a reduction of the signal intensity (Fig. 2a)[9]. $T_1$, $T_2$, and the decoherence time, $T_2^* \equiv (1/T_2 - 1/2T_1)^{-1}$, are plotted for two representative samples biased at $-1.5 \, V$ and $-0.4 \, V$ in Fig. 2b and c, respectively, while a third is shown in Supplementary Note 3 for $-1.0 \, V$. Relaxation times were only weakly temperature-dependent and remarkably long—on the order of microseconds across the entire temperature range, including at room temperature. In all samples, $T_2$ was slightly shorter than $T_1$ and increased monotonically with increasing temperature. $T_1$ exhibited a similar temperature dependence, except at very low temperatures and near room temperature, where it may have decreased slightly with increasing temperature.

This behavior differs starkly from our recent FI-ESR studies on high-mobility conjugated polymers[9] and from studies on C10-DNBDT-NW[15]. In our work on conjugated polymers, we observed orders-of-

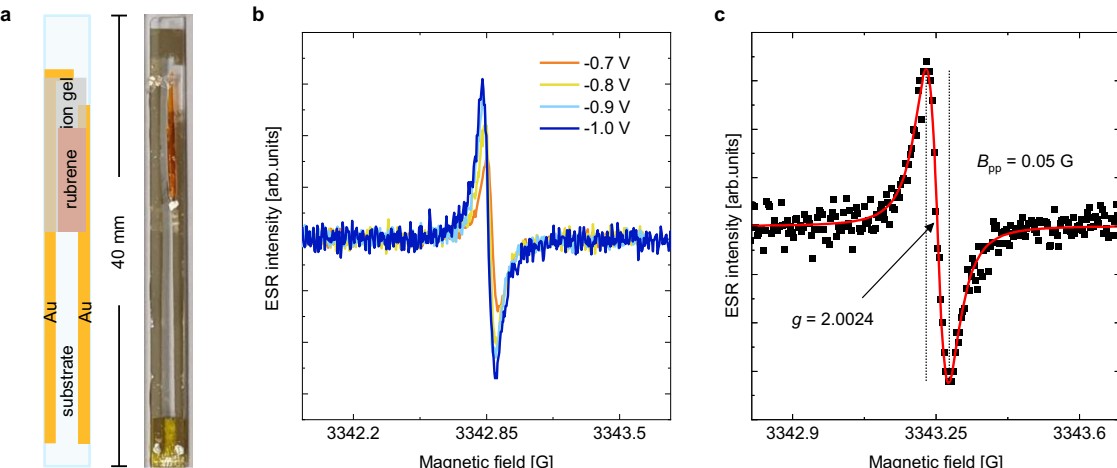

**Fig. 1 | Characteristic FI-ESR curves on ion-gel-gated rubrene. a** A schematic of the capacitor architecture (left) and an image of an actual device (right). In the schematic, the gold rectangles are electrodes, the orange rectangle rubrene, the dark-gray rectangle the ion gel, and the large, translucent blue rectangle the substrate. **b** The evolution of the observed ESR signal with increasing gate voltage. **c** A representative ESR spectrum at bias voltage − 1.2 V and temperature 290 K for our ion-gel-gated rubrene devices. The black dots are raw data points, and the red curve is the fitting result.

magnitude changes in $T_1$ and $T_2$ with temperature, with $T_1$ decreasing monotonically with temperature across the entire range and approaching one microsecond only at low temperatures, and with $T_2$ exhibiting motional narrowing only between 100 and 200 K and otherwise decreasing monotonically with temperature. In C10-DNBDT-NW, where momentum-scattering processes drive relaxation, it was observed that relaxation times decreased monotonically with temperature over the range measured. Compared to these systems, the temperature-dependent relaxation behavior in rubrene is remarkable, and results in room-temperature spin lifetimes that are 10–100 times longer than in polymers and in C10-DNBDT-NW, and 1000 times longer than in many inorganic systems[37–39] and graphene[40].

Though the general increase in $T_2$ with temperature is a well-known effect of motional narrowing in systems with thermally activated charge transport, the corresponding increase in $T_1$ has not been observed before in organic semiconductors. To explain it, we use the Redfield theory of spin relaxation and motional narrowing[22]. In this framework, relaxation is driven by fluctuations of the local fields $\mathscr{B}_x$, $\mathscr{B}_y$, and $\mathscr{B}_z$, which are correlated with themselves on timescale $\tau_c$. $T_1$ relaxation results from fluctuations of the transverse components ($\overline{\mathscr{B}_x^2}$ and $\overline{\mathscr{B}_y^2}$) at the Larmor frequency $\omega_L$ because they are able to induce spin-flips; $T_2^*$ relaxation is driven by fluctuations in $\overline{\mathscr{B}_z^2}$ because they produce spatial variations in $\omega_L$ and thus drive decoherence; and $T_2$ relaxation includes the lifetime broadening acquired from both spin-flips and from decoherence. The equations for the relaxation times are

$$
\begin{aligned}
\frac{1}{T_1} &= \gamma_e^2 \left( \overline{\mathscr{B}_x^2} + \overline{\mathscr{B}_y^2} \right) \frac{\tau_c}{1 + \omega_L^2 \tau_c^2} \\
\frac{1}{T_2} &= \frac{1}{2T_1} + \gamma_e^2 \overline{\mathscr{B}_z^2} \tau_c \\
\frac{1}{T_2^*} &\equiv \frac{1}{T_2} - \frac{1}{2T_1} = \gamma_e^2 \overline{\mathscr{B}_z^2} \tau_c,
\end{aligned}
\tag{1}
$$

where $\gamma_e$ is the free-electron gyromagnetic ratio. In the case of random charge motion with the fluctuating fields dominated by hyperfine interactions, we can assume that the magnitude of the fluctuating fields are isotropic: $\overline{\mathscr{B}_{rms}^2} = \overline{\mathscr{B}_x^2} = \overline{\mathscr{B}_y^2} = \overline{\mathscr{B}_z^2}$[22]. Figure 2d illustrates schematically how $T_1$ and $T_2$ are expected to vary with $\tau_c$ (normalized by $\omega_L$). $T_1$ relaxation is most effective when the correlation time matches the Larmor period, and becomes less effective as $\tau_c$ increases or decreases; this is simply because spin-flips require an exchange of

energy with the lattice via a photon of energy $\omega_L$. $T_2$, on the other hand, can be relaxed by spin-flips or by pure decoherence. Motional narrowing occurs when faster charge motion reduces $\tau_c$, allowing each spin to average over different magnetic environments and thus reducing the rate of decoherence. In this regime, the charge hopping frequency $\nu$ can be interpreted as the inverse correlation time[9].

For systems with thermally activated transport, an increase in temperature corresponds to moving from right to left in Fig. 2d. In our previous experiments on thermally activated conjugated polymers[9], we were in the regime to the right of the minimum, where we observed a decrease in $T_1$ and an increase in $T_2$ with increasing temperature. Here, in our ion-gel-gated rubrene crystals, charge transport is also temperature-activated (see discussion below)[41], but the microscopic charge motion leading to motional narrowing is sufficiently fast that we enter the regime to the left of the minimum, where both $T_1$ and $T_2$ increase as charge motion becomes faster at higher temperatures.

To examine how well quantitatively the Redfield equations reproduce the observed behavior, we require a functional form of the charge motion. For this, as discussed in Supplementary Note 3, we express the charge motion as a temperature-activated process super-imposed onto a constant (or comparatively weakly temperature-dependent) background process:

$$
\tau_c^{-1}(T) = \nu(T) \propto A \times T \exp\left[ \frac{-E_A}{k_B T} \right] + C
\tag{2}
$$

The temperature-independent process represents a contribution from the fast local motion of carriers due to transient localization within local regions defined by ionic potentials, which is assumed here to be much more independent of temperature than the temperature-activated process that represents the slower charge hopping between these regions. In Supplementary Note 3 we discuss a simple method for fitting the $T_2^*$ values deduced from experiment as a function of temperature to Equation (2). From this, we deduce a best-fit activation energy of 40 meV, which is consistent with measurements of the temperature-dependent mobility of rubrene electric double-layer transistors (EDLTs)[42]. However, while the $T_2^*$ data is sufficient to fit $E_A$, it is not possible to determine $\tau_c$ as a function of temperature by this method. This is because $\overline{\mathscr{B}_{rms}^2} \tau_c \propto 1/T_2^*$, therefore, scaling $\tau_c$ together with $\overline{\mathscr{B}_{rms}}$ such that the product $\overline{\mathscr{B}_{rms}^2} \tau_c$ is held constant and

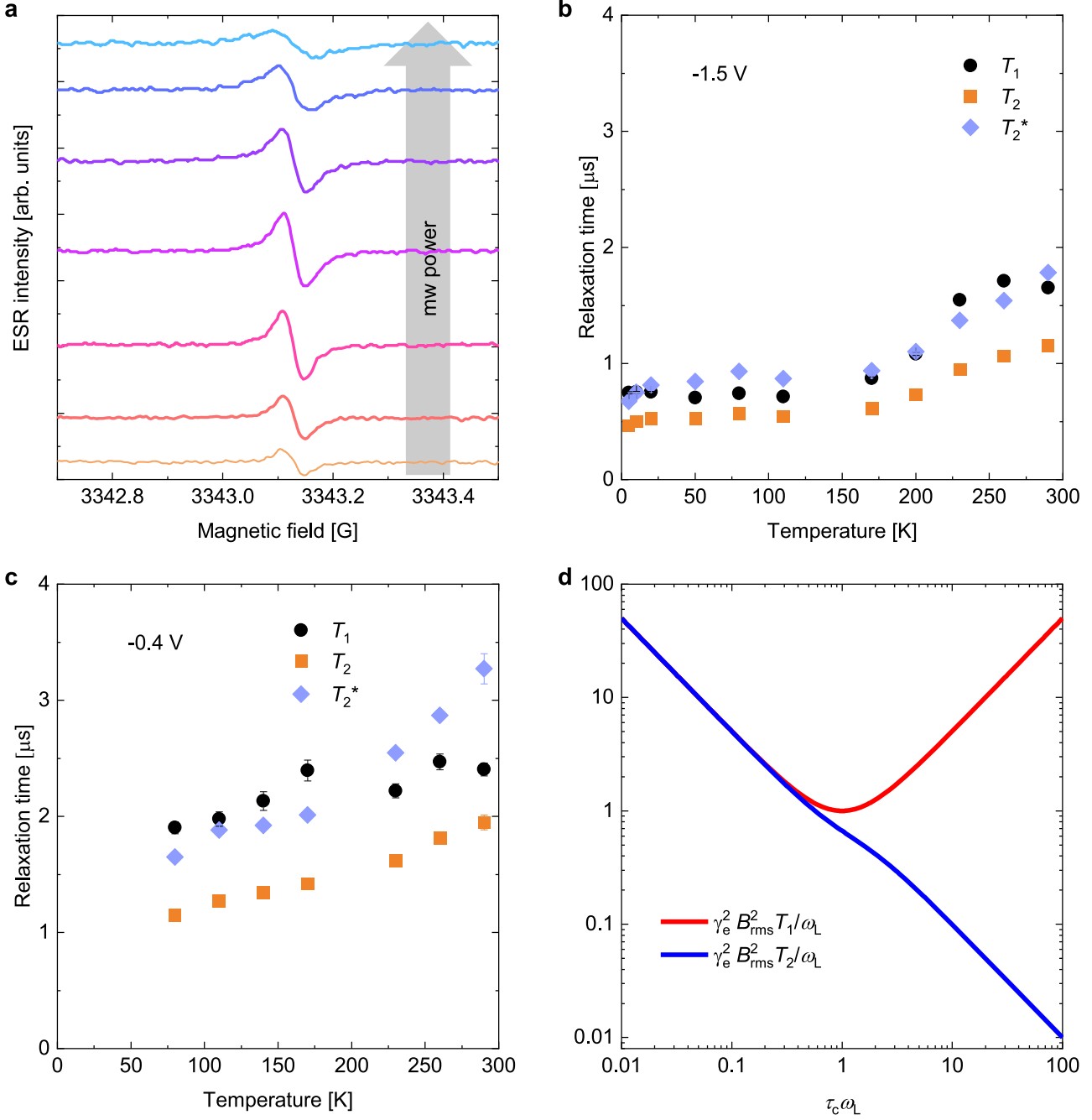

**Fig. 2 | Relaxation times in ion-gel-gated rubrene devices. a** Power saturation measurements recorded at −1.0 V and 290 K. The extracted $T_1$, $T_2$, and $T_2^*$ are plotted in (**b**) for a rubrene ion-gel-gated capacitor at −1.5 V and in (**c**) for a second device at −0.4 V. **d** Normalized $T_1$ ($\gamma_e^2 B_{rms}^2 T_1/\omega_L$) and $T_2$ ($\gamma_e^2 B_{rms}^2 T_1/\omega_L$) as a function of correlation time in the motional narrowing regime according to Equation (1)

(Adapted from Reference[22]). The absence of the − 0.4 V data at $T < 80$ K is due to the smaller number of polarons in the accumulation layer and the shorter relaxation times at low temperatures, resulting in signals that are unobservable or could not be fit with our techniques. Error bars represent standard deviations.

does not lead to changes to $T_2^*$ or the spectral lineshape in the motional narrowing regime.

To determine $\overline{\mathscr{B}_{rms}}$ and $\tau_c$ uniquely, we fit $T_1$ and $T_2$ using Equations (1) and (2) for varying values of $\overline{\mathscr{B}_{rms}}$. At lower charge density ($V_g = − 0.4$ V, Supplementary Fig. 5), we can self-consistently fit $T_1$, $T_2$, and $T_2^*$ with $\overline{\mathscr{B}_{rms}} = 0.93$ mT, similar to previous measurements[43]. This $\overline{\mathscr{B}_{rms}}$ value implies a correlation time varying from 10 - 20 ps between 0 K and room temperature (Supplementary Fig. 5d). At higher charge densities ($V_g = − 1.5$ V), the correlation time becomes sufficiently short that we remain in the motional narrowing regime even at low

temperature. However, in this regime relaxation is also affected by dipolar coupling, as discussed in detail below, and we are unable to self-consistently fit the data using an isotropic $\overline{\mathscr{B}_{rms}}$. Further discussion of this regime is given in Supplementary Note 4.

Our results demonstrate that in our rubrene crystals, the time-scale for charge motion is faster than that of the Larmor frequency ($\omega_L \approx (17$ ps$)^{-1}$). This implies that the local mobility of holes in rubrene must be at least 100 times higher (~ 10 cm² V⁻¹s⁻¹) than in polymers like indacenodithiophene-co-benzothiadiazole (IDT-BT)[9], for which the motion frequency was estimated to be 1 GHz, i.e., two orders of magnitude slower than $\omega_L$. While we do not record such high mobilities on

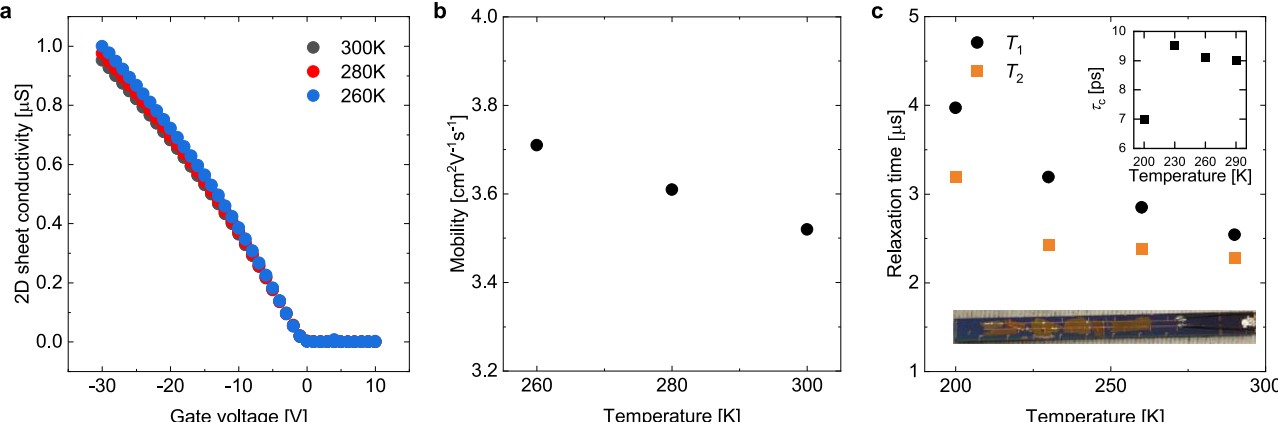

**Fig. 3 | Charge-transport and FI-ESR data of a conventional rubrene FET.**
**a** Conductivity and (**b**) corresponding mobility data at 260, 280, and 300 K. The temperature-inhibited mobility confirms that charge transport in this device is dictated by transient localization. **c** $T_1$ and $T_2$ in the device from 200 to 290 K. The microsecond-long relaxation times are nearly the same as those observed in our

ion-gel-gated devices in this temperature regime. This supports our interpretation that transient localization sets the overall magnitude of the relaxation times for rubrene devices. Insets: an image of a conventional rubrene ESR device with multiple crystals (bottom) and the corresponding correlation time $\tau_c$ from 200 to 290 K (top).

rubrene EDLTs (see Supplementary Note 5), such values are not unreasonable at the local scale, where charge motion should be similar to that of rubrene single crystals without ions[32,44–48]. The mobility of EDLTs reflects long-range transport, which is influenced by electrostatic potential fluctuations created by the ions and may greatly underestimate the local transport driven by transient localization.

Comparing to C10-DNBDT-NW[15], we note that C10-DNBDT-NW contains relatively heavy sulfur atoms and that Elliot-Yafet scattering is driven by spin-orbit interactions. In our ion-gel-gated rubrene, not only is the strength of spin-orbit coupling weaker, but the presence of long-range electrostatic potentials means we do not enter the delocalized, band-like transport regime required for Elliot-Yafet scattering. Our interpretation is that the very weak spin relaxation in our systems reflects effective motional narrowing induced by fast transient-localization motion of the carriers within the local regions to which motion is confined by the long-range electrostatic potential fluctuations induced by the ions. The main effect of temperature is that it allows charges to leave these local regions, therefore making motional narrowing even more effective. This interpretation is consistent with the lifetimes in Fig. 2b becoming constant and remaining long at temperatures below 100 K.

## Comparison to an FET with pure transient localization

The above picture of fast local transient-localization motion within regions defined by the ionic potentials is plausible, but it is a hypothesis that should be validated. For this, we performed a series of electrical and FI-ESR measurements on rubrene FETs gated with $SiO_2$ and Cytop, which is a more conventional design (see "Methods" and Supplementary Note 6). This is a very challenging experiment, since the number of spins that can be induced by the field effect in one individual single crystal is not high enough to achieve a good ESR signal-to-noise ratio. We therefore fabricated FI-ESR devices in which multiple crystals were integrated into the cavity. Due to the absence of the ion gel, this architecture excludes temperature-activated hopping between potential wells and allows us to probe in a direct manner the dynamics due to transient localization. We confirmed that this is indeed the case by measuring the 2D sheet conductivity vs. gate voltage and extracted mobility vs. temperature data, the plots for which are shown in Fig. 3a and b, respectively. The decrease of mobility with increasing temperature is a clear indication of the band-like behavior predicted by transient localization. (The electrical characterization of a second device is shown in Supplementary Note 7; it displays the same band-like dependence on temperature). The price to pay in this

challenging experiment was that we were only able to access a more restricted temperature range between 200 K and room temperature before we lost electrical contact to some of the crystals due to problems with mismatched thermal expansion.

Corresponding relaxation times for a conventional FET are shown in Fig. 3c at 200, 230, 260, and 290 K, with its optical image shown in the inset. Across the temperature range, we again observed microsecond-long relaxation times, a clear indication that the overall magnitude of the spin lifetimes are set by transient localization. In contrast to our ion-gel-gated devices, however, here we observed the opposite temperature dependence, i.e., shortening spin lifetimes with increasing temperature. This is expected for a device in which only transient localization is active and in good agreement with the observed temperature dependence of the mobility (Fig. 3b): as temperature increases, so does dynamic disorder, which results in slower motion, lower mobilities, and consequently an increase of the correlation time. In the ion-gel-gated devices, the observed increase in relaxation times with temperature is due to an additional, temperature-activated charge-transport mechanism at play; even though increased temperature hampers transient localization and therefore charge motion, it facilitates hopping between ionic potentials, which leads to overall faster charge motion. We can use the value of $\mathscr{B}_{rms}$ extracted above (0.93 mT) to estimate the correlation time $\tau_c$ from the conventional FET data to be in the range of 7-9 ps between 200 K to 290 K (inset of Fig. 3c). These provide reliable values for the experimentally determined timescale for charge motion on the surface of the crystals in the absence of any perturbing ion potentials that can be directly compared to predictions of theoretical models.

Transient-localization physics provides the most appropriate framework for interpreting the experimentally determined correlation times in terms of the underpinning motion of the charge carriers. We therefore performed simulations of the motion of holes in rubrene crystals in the transient localization regime using the fragment-orbital based surface hopping (FOB-SH) method, which couples a quantum mechanical simulation of the electron dynamics to a classical simulation of the structural, molecular dynamics[2]. These simulations are essentially identical to those in ref. 2, but were performed over the longer timescale of 10 picoseconds to approach more closely the timescales probed in the experiments (see "Methods"). As discussed above, a characteristic feature of the transient localization regime is that the hole wavefunction undergoes rapid fluctuations between localized and delocalized configurations, which arise from the coupling of the electrons to the molecular dynamics. They can be

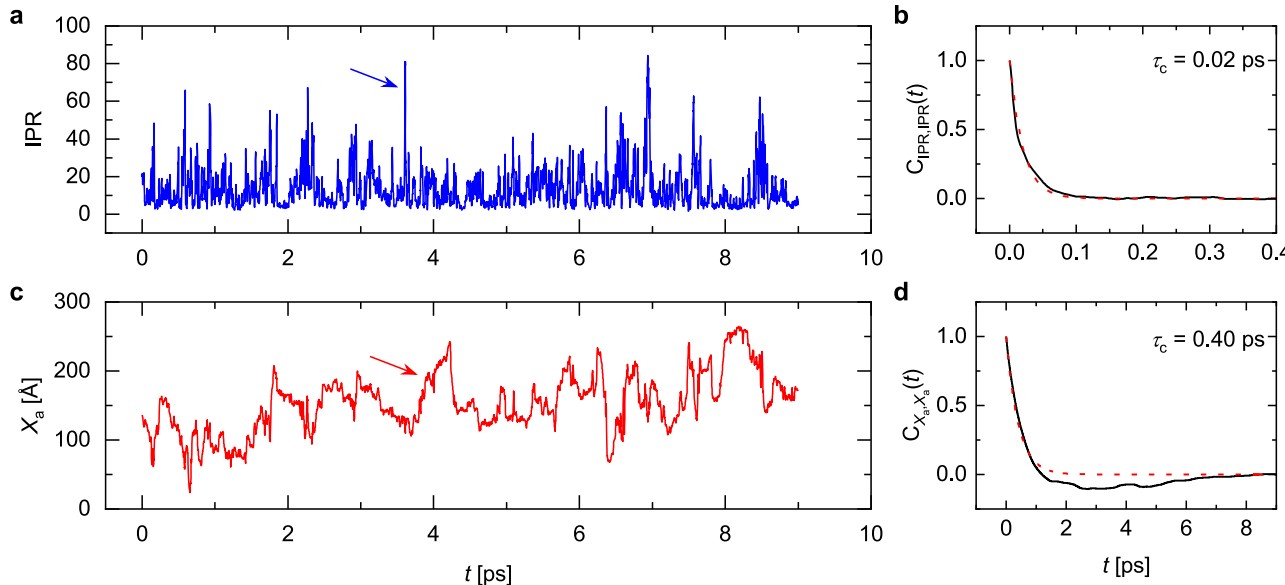

**Fig. 4 | Simulated transient-localization motion in rubrene.** The inverse participation ratio (IPR, panel **a**) and the displacement of the center of charge of the hole wavefunction (panel **c**), shown along a single representative FOB-SH non-adiabatic dynamics trajectory. Notice the large delocalization and corresponding displacement marked by an arrow. The auto-correlation functions for IPR and displacements were averaged over 50 FOB-SH runs and are shown in panels (**b**) and (**d**), respectively (black lines). The correlation times indicated were obtained from the best mono-exponential fit functions (red dashed lines).

visualized by plotting the inverse participation ratio (IPR), which is a measure of the number of molecules $N$ over which the carrier wavefunction is delocalized, as a function of time (Fig. 4a). These are expected to be one source of the magnetic field fluctuations relevant for the ESR interpretation, because the magnetic field experienced by a carrier delocalized over $N$ molecules is essentially an average over the randomly oriented hyperfine and spin-orbit fields generated by each molecule and should therefore scale as $\frac{1}{\sqrt{N}}$. However, the fluctuations in carrier delocalization occur on the very fast timescale of about 20 femtoseconds, which can be estimated by fitting an exponential decay function to the IPR($t$) autocorrelation function (Fig. 4b). This is much faster than the correlation time deduced from the experiments and is therefore likely to provide only a minor contribution to the observed spin relaxation.

However, there is a second source of magnetic field fluctuations expected within the transient localization framework. The rapid fluctuations in IPR occasionally result in highly delocalized wavefunction configurations (arrow in Fig. 4a); there is a chance that when the wavefunction returns to a more localized configuration after each such delocalization event, its center-of-mass position, $X_a(t)$, shifts significantly. These long-distance jumps (indicated by the arrow in Fig. 4c), which govern the charge carrier mobility, transport the wavefunction to a new region in the crystal, where the carrier experiences a different magnetic field environment. They occur on a timescale of about 400 femtoseconds (as estimated from the $X_a(t)$ autocorrelation function (Fig. 4d)), which approaches the experimentally observed correlation times, but is still about one order of magnitude faster. However, this remaining discrepancy between experiment and theory is likely to be explained mostly by the fact that the theoretical simulations were performed on a perfect single crystal with a simulated room-temperature mobility greater than 30 cm²V⁻¹s⁻¹ and in the absence of any static disorder and trap states. This mobility is 10 times higher than the mobilities of our cytop-gated rubrene FETs, and even higher than those of the best-performing rubrene air-gap FETs (15–20 cm² V⁻¹s⁻¹)[32,44–46,48]. The fact that in the simulations we neglect that charge carriers spend some of their time in localized trap states in between periods of transient-localization-mediated diffusive motion is likely to be the main reason for the simulations

underestimating the correlation time by about one order of magnitude. This consideration leaves limited room for contributions from other spin-relaxation mechanisms, such as the spin-orbit-interaction-mediated relaxation that has been observed in organic semiconductors comprising heavier elements[9,15]. We believe that it would be possible to construct a quantitative model that accurately calculates the magnetic field experienced by the carrier wavefunction at each point in time and also takes into account the effect of charge-carrier trapping. It has been demonstrated that within FOB-SH it is possible to build realistic models of disorder in pentacene[49]; to build such a realistic model for rubrene crystals and use it to model the FI-ESR data goes beyond the scope of the present work, but will be attempted in the future. In the present work, we have demonstrated clearly that, in principle, the combination of ESR experiments with simulations of transient localization provides a powerful approach for directly probing the motion of charge carriers in molecular crystals.

## Spin-spin dipolar interactions at high carrier densities

Finally, we turn to another interesting question that can be addressed in such FI-ESR experiments: estimating carrier concentration in the accumulation layer at the interface, where carriers stop behaving as non-interacting electrons and instead begin to couple to one another. This should be detectable in an FI-ESR experiment through the onset of an additional spin relaxation mechanism due to spin-spin interactions. In conventional field-effect-gated devices, charge densities are too low to observe this phenomenon, i.e., there is no change in ESR lineshape up to the highest applied gate voltage. However, in our ion-gel-gated devices, we observed that we were able to induce much higher charge concentrations. Our analysis in the previous sections suggests that ion-gel devices present an environment in which spin lifetimes are lengthened by hopping between ionic potential wells. Such hopping not only allows spins to sample different magnetic environments, but also presents opportunities for them to interact with one another. Such interactions could shorten spin lifetimes under the right circumstances, such as what occurs in dipolar broadening. To probe the possibility of detecting such interactions, we performed a series of measurements on ion-gel-gated devices as a function of carrier density near room temperature. Figure 5 shows plots of the transverse

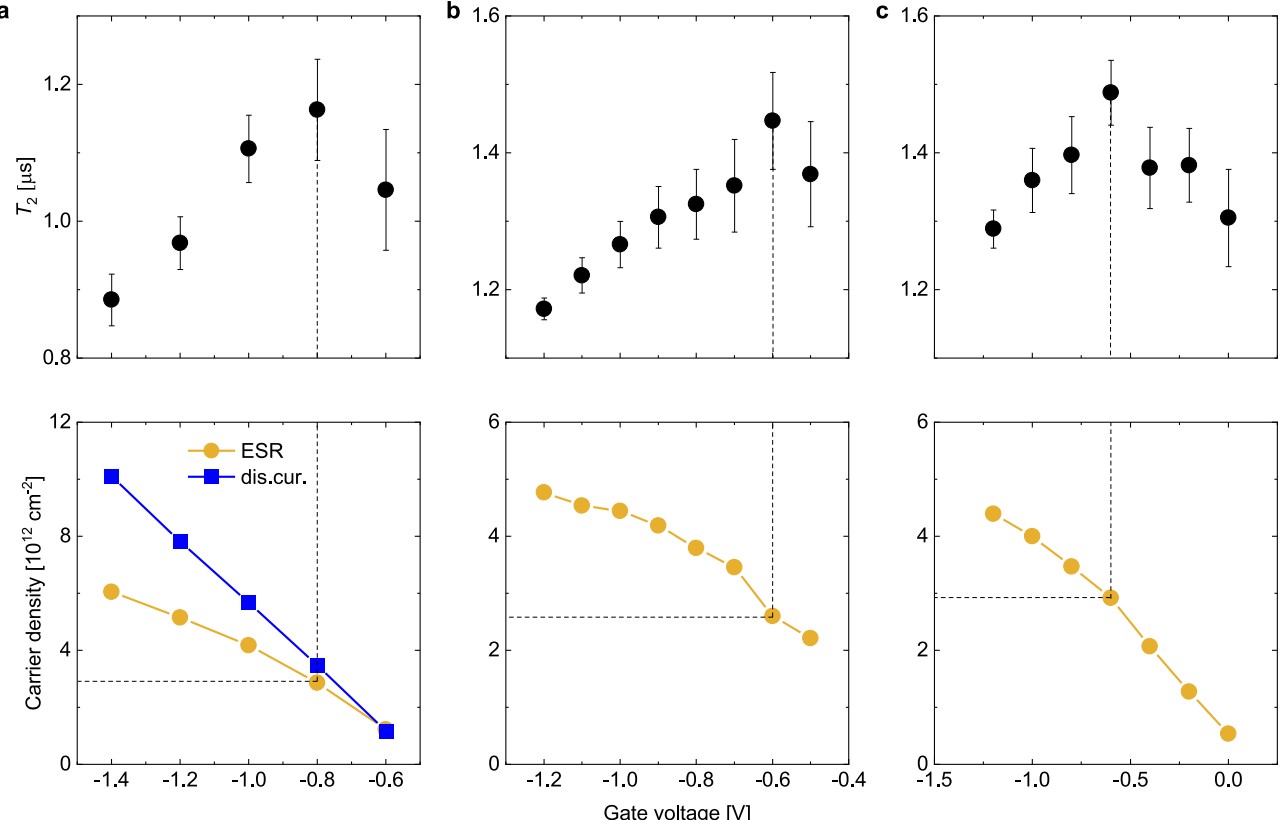

**Fig. 5 | $T_2$ and carrier density vs. gate voltage in three devices.** Because all lineshapes were Lorentzian and the signal was not saturated, $T_2$ was calculated from the inverse of the half-width-at-half-height of the ESR signal, $\Delta B_{1/2}$. For Device 1 (**a**), both the ESR-extracted carrier concentration (ESR, yellow circle) and that of the displacement current (dis.cur., blue square) are shown. For Devices 2 (**b**) and 3 (**c**), only the ESR-extracted carrier concentration is shown. In all devices, a maximum in $T_2$ occurs at an area concentration of about $3 \times 10^{12}$ spins cm$^{-2}$. Error bars represent standard deviations.

relaxation time as a function of gate voltage (and therefore carrier concentration) for three different devices. As shown, we recorded a small but distinct peak in $T_2$ as a function of gate voltage that is reproducible and outside the bounds of measurement error. In the lower panels of Fig. 5a–c, we show the number of spins calculated from ESR by doubly integrating the measured first-derivative signal. For Device 1 (Fig. 5a) we also plot an estimate of injected carrier concentration from the measured displacement current (see Supplementary Note 8). The peak in $T_2$ corresponds to a carrier concentration of 2.8, 2.6, and $2.9 \times 10^{12}$ cm$^{-2}$ in Device 1, 2, and 3, respectively. In addition, the weak gate-voltage-dependence of $T_2$ further indicates that Rashba-type spin-orbit interaction does not play an important role, as expected from the structure of rubrene (see Supplementary Note 9 for a discussion on the Rashba effect).

Before developing a model to explain these results based on spin-spin interactions, it is important to exclude the possibility that this gate-voltage dependence of $T_2$ could be explained by a corresponding peak in the hopping frequency (and therefore modeled by the Redfield equations). To do so, we performed ESR measurements on a rubrene crystal EDLT (Supplementary Note 5) fabricated under conditions similar to those of the capacitance devices. In the EDLT, we observed a peak in $T_2$ at a very similar carrier concentration of $2.5 \times 10^{12}$ cm$^{-2}$ (Supplementary Fig. 7a), yet the corresponding EDLT transfer characteristics (Supplementary Fig. 7b) showed no peak in the drain current $I_{sd}$ or the extracted mobility around this value, thus excluding an increase in the hopping frequency as an explanation. We note that a peak in conductance and mobility has been observed in similar EDLT devices previously, but only at carrier concentrations significantly higher than $10^{13}$ cm$^{-2}$ [41,50].

To validate our interpretation of the peak in $T_2$ as the onset of spin-spin dipolar interactions, we now consider a model initially developed by Abragam[51] for rigid spheres in fluid NMR and crudely adapted here for electronic spins in field-induced ESR. We begin with the perturbing Hamiltonian. Instead of writing the classical dipolar interaction in Cartesian coordinates, we note that, most generally, any spin-spin interaction can be expressed in terms of two operators: one that acts on the relative position of the two spins involved, and a second that acts on their spin components. Because we are concerned with interactions between two spins, these two operators are tensors of the second rank. This allows us to label the lattice operator $F_k^{(2)}$ and the spin operator $\mathcal{S}_k^{(2)}$, where we have chosen to work in spherical harmonic coordinates rather than Cartesian ones, meaning the index $k$ runs over linear combinations of the Cartesian components. The perturbing Hamiltonian is then

$$\mathcal{H}_1 = \sum_k F_k^{(2)} \mathcal{S}_k^{(2)}, \tag{3}$$

and the degree to which the position function $F_k^{(2)}$ is correlated with itself on timescales of $\tau$ is determined by the autocorrelation function

$$\overline{F_k^{(2)}(t) F_{k'}^{(2)}(t+\tau)} = \delta_{kk'} G_k^{(2)}(\tau). \tag{4}$$

We also define the corresponding spectral density function

$$J_k^{(2)}(\omega) = \int_{-\infty}^{\infty} G_k^{(2)}(\tau) e^{-i\omega\tau} d\tau. \tag{5}$$

Abragam showed that, after evaluating over the spin coordinates, the longitudinal and transverse relaxation times are directly proportional to the different $k$ components of the spectral density function, Equation (5). Thus, to evaluate them, a form must be chosen for the position function $F_k^{(2)}$. Abragam considered the case of liquid and gas nuclear magnetic resonance, wherein spin-carrying molecules translate past each other and undergo a diffusive process described by the diffusion current density **j**,

$$\mathbf{j} = qD\nabla n, \tag{6}$$

where $q$ is the fundamental electric charge, $n$ is the carrier density, and $D$ the diffusion constant. For our analysis, we note that the diffusion coefficient is that of charge carriers/spins and can be expressed in terms of the carrier mobility using the Einstein relation, $D = \mu k_B T/q$. Based on our measurements of the bulk mobility in the EDLT, we estimate the hole mobility to be between 0.01 and 0.1 cm$^2$ V$^{-1}$s$^{-1}$, a low value we attribute in our devices (compared to the work of Xie and Frisbie[50]) to an increased access resistance from the crystal thickness due to the top-gate, bottom-contact architecture used.

Abragam's analysis assumed a minimum distance of approach $d$ equal to twice the radius of the spheres. Here, we crudely take this minimum distance to be defined by the spatial extent of the hole wavefunction, i.e., we assert that two holes may touch, but may not overlap. From the FOB-SH simulations discussed above, we can estimate the 1D charge delocalization length (and thus minimum distance of approach) in rubrene to be approximately 5-6 nm. The spectral density, Equation (5), consequently loses its dependence on $\omega$ because it is so small[51], and we thus have for the relaxation times

$$\frac{1}{T_1} = \frac{3}{2}\frac{\mu_0^2}{16\pi^2}\gamma^4\hbar^2 S(S+1)\frac{16\pi}{45}\frac{n}{dD}$$
$$\frac{1}{T_2} = \frac{\mu_0^2}{16\pi^2}\gamma^4\hbar^2 S(S+1)\frac{8\pi}{15}\frac{n}{dD}. \tag{7}$$

Using values of $n = 2.8 \times 10^{19}$ spins cm$^{-3}$, $d = 5$ nm and mobility of 0.01-0.1 cm$^2$ V$^{-1}$s$^{-1}$ to estimate $D$ from the Einstein relation, we estimate $T_2 = 33-330$ μs (for the validity of the classical Einstein relation and the non-degenerate nature of the system, see Supplementary Note 10). We acknowledge that this simple model overestimates $T_2$ by 1 - 2 orders of magnitude, and a more refined model will need to consider several additional factors. For example, estimating the minimum distance of approach is complicated by the rapid temporal fluctuations of the charge wavefunctions that are characteristic of the transient localization regime, but neglected here. Furthermore, the diffusion in our system is restricted to two dimensions because it occurs only in the accumulation layer, which is likely to result in an underestimate of the rate at which spins experience close encounters. We are also aware that charge and spin can exhibit different diffusion coefficients[52,53]. In certain cases, spin-diffusion coefficients can exceed charge-diffusion coefficients in organic semiconductors due to enhanced exchange coupling[53]. However, it's unclear if this applies to rubrene, and using a larger spin-diffusion coefficient in our model would lead to even greater deviations from the experimental results. Despite these shortcomings, however, the model does nicely explain the presence of a peak in $T_2$ as the interplay between the gate-voltage-dependence of the mobility and the carrier concentration.

It remains to justify the use of the relatively low EDLT mobility here to explain relaxation driven by dipolar interactions compared to the use of the (higher) local mobility used in the Redfield equations to explain motional narrowing: When considering relaxation driven by local fluctuations due to the hyperfine interaction, a charge need not leave its local region. Within a single potential well, an electron can sample many different hyperfine fields by means of transient localization. Therefore, motional narrowing is driven by this fast charge motion. However, for relaxation driven by dipolar coupling, a charge

must leave its local potential region to interact with a neighboring spin. In this case, it must be the mobility determined by inter-ionic-potential-well hopping that is relevant.

## Discussion

We have experimentally identified a mechanism for achieving long spin lifetimes in semiconductors at room temperature, which relies on the combination of weak spin-orbit coupling and high carrier mobility. Weak spin-orbit coupling minimizes the impact of momentum scattering on spin relaxation, while high carrier mobility extends spin lifetimes by averaging out local hyperfine fluctuations. Our analysis within the transient localization framework shows that by optimizing crystal quality and reducing the influence of charge carrier trapping, it might be possible to achieve even longer spin relaxation times in rubrene, potentially by an order of magnitude. We believe that the combination of weak spin-orbit coupling and high carrier mobility could serve as a guideline for developing materials with long spin lifetimes, extending beyond the realm of organic semiconductors.

Our work provides valuable insights into the spin-relaxation physics of high-mobility molecular crystals with low spin-orbit coupling. It represents the observation of a spin-relaxation regime in organic semiconductors wherein the fast, local transient-localization motion of spins gives rise to effective motional narrowing. In this regime, charge motion occurs faster than the Larmor frequency, and the spin relaxation times $T_1$ and $T_2$ increase in tandem as charge motion becomes faster. Therefore, microsecond-long spin relaxation times are achievable, even at room temperature. We have also demonstrated that the combination of FI-ESR experiments and transient localization simulations provides a powerful approach for directly probing charge carrier motion in the unique charge transport regime that is found in such molecular crystals.

## Methods

### Rubrene crystal growth

Rubrene single crystals were grown by the physical vapor transport method[54] using a horizontal tube furnace. Rubrene powders (Sigma-Aldrich) were placed in a ceramic boat that was then loaded into a quartz tube and placed at the center of the furnace (the hottest region). With a furnace temperature of 290 °C and an argon gas flow rate of 50 sccm, rubrene single crystals were collected after a growth time of 10–20 h. Only those crystals with molecularly smooth surfaces were used for device fabrication.

### Ion-gel-gated device design for FI-ESR

Figure 1a shows a schematic of the device structure used. Two parallel gold electrodes (gold rectangles) were evaporated onto one side of a $40 \times 3$ mm$^2$ fused-quartz plate (UQG Optics, FQP-5005) (translucent blue-gray rectangle). A single rubrene crystal (orange rectangle) of approximate dimensions $10 \times 1$ mm$^2$ was gently placed onto the glass so as to overlay with one of the electrodes, then affixed with a small drop of silver paint. The ion-gel solution was prepared by mixing poly(vinylidene fluoride-co-hexafluoropropylene) (PVDF-HFP), ionic liquid 1-butyl-1-methylpyrrolidinium bis(trifluoromethylsulfonyl)imide ([BMP][TFSI]), and acetone with a weight ratio of 1:4:20, and the ion-gel films were drop-cast onto glass substrates. A thin slice of ion gel film (translucent gray rectangle in Fig. 1a) was then transferred onto the device so that it connected the top surface of the rubrene to the second electrode. This created a capacitor architecture. We chose this design because it allowed us to measured a single, as-large-as-possible rubrene crystal inside the cavity and protected the devices from degradation under thermal stress. Of course, it presented the disadvantage that we were unable to directly measure FET mobilities during the ESR measurements (difficulties in performing ion-gel-gated transistor and ESR measurements in one identical rubrene crystal can be found in Supplementary Note 11).

An actual device that was measured is also shown in Fig. 1a. The sample is attached to the sample holder via double-sided Kapton tape. The three contact strips on the holder are clearly visible, as are the soldering joints that affix the wires that lead to the Keithley 2612B Source-Measure Unit. Note that, due to the device architecture, only two of the three contact pads were used; these were wire-bonded to the two electrodes on the device. The silver dag was used to prevent the wire from breaking off (due to thermal strain) the contact pads and/or the electrodes. The small piece of silver dag on the rubrene was needed to adhere the rubrene crystal to the substrate because the adhesion between only the substrate and the large rubrene crystal was not sufficient.

### FI-ESR measurements

The device was loaded into the spectrometer setup as follows: After affixing the sample to the sample holder, the contact pads of the sample were wire-bonded to the contact strips of the holder using aluminum thread. The sample-and-holder combination was then slid into a Wilmad Suprasil EPR tube (Sigma-Aldrich, product no. Z5674XX) and sealed under nitrogen. The sample was then loaded into an Oxford Instruments ESR900 cryostat, which was controlled by an Oxford Instruments Mercury iTC. ESR measurements were taken on a Bruker E500 spectrometer using a Bruker ER 4122SHQE cavity and an X-band microwave source, and a Keithley 2612B Source-Measure Unit was used for electrical characterization. The direction of the applied magnetic field was perpendicular to the device plane. CustomXepr, a Python package developed by Sam Schott, was used to integrate the above-mentioned instruments and automate measurements when desired[9].

Initial measurements showed a peak-to-peak linewidth of approximately 0.05 G, meaning artifact-less spectra could only be obtained by measuring with a modulation amplitude of 0.02 G and a modulation frequency of 50 kHz[55]. Because the injection efficiency of ion gels varies with sample dimensions, the minimum voltage at which a signal could be observed varied across samples. The range of observed minimum values was $+ 0.4$ V to $- 0.6$ V. For low-temperature ESR measurements, the gate voltage was applied at room temperature, and then the device was cooled down with the gate voltage applied.

### Conventional FET design

Conventional rubene FETs were fabricated on undoped silicon wafers with a 300 nm $SiO_2$ layer to minimize noise in the ESR signal from the substrate. To get a better semiconductor/dielectric interface, a thin layer of Cytop (40–45 nm) was spin-coated on top of the $SiO_2$ from a diluted solution (Cytop CTL-809M: solvent = 1: 5, AGC Inc.) at 2000 rpm for 1 min and annealed at 120 °C for 30 min. (The reason for choosing this device geometry can be found in Supplementary Note 6.) The resulting dielectric layer had a specific capacitance of 9 nF cm$^{-2}$. Au source/drain electrodes (20 nm) were thermally evaporated on top of Cytop through a shadow mask. Finally, thin rubrene crystals were manually laminated over the channel to complete the fabrication of the FETs. For FI-ESR devices, multiple crystals were laminated on the same device to maximize the channel area in order to get a spin number much larger than the detection limit of our ESR setup.

### Fragment orbital-based surface hopping simulations

Fragment orbital-based surface hopping (FOB-SH) is a mixed quantum-classical dynamics method based on the fewest switches surface hopping that enables efficient simulation of charge transport in organic semiconductors on the nanoscale[2,56]. A comprehensive overview of the methodology can be found in References[57–59] and details on the force field and electronic Hamiltonian parameterization for rubrene can be found in Reference[60]. We used a supercell of $38 \times 10 \times 1$ unit cells (corresponding to a high-mobility plane of 27.4 nm × 14.5 nm), where the first dimension corresponds to the high-mobility direction $a$. The supercell was equilibrated to 300 K for 200 ps using

classical molecular dynamics in the NVT ensemble with a Nosé-Hoover thermostat and a MD time step of 1 fs. Initial phase space coordinates for 50 FOB-SH trajectories were obtained from uncorrelated snapshots taken every 0.5 ps along a subsequent NVE trajectory. For each trajectory, the wavefunction was initialized on a single molecular site at the corner of the simulation box and propagated for 10 ps using a MD time step of 0.1 fs and an electronic time step of 0.02 fs for integration of the time-dependent electronic Schrödinger equation with the 4th order Runge-Kutta algorithm. The initial 1 ps of dynamics was neglected in all analysis. The delocalization of the hole wavefunction is quantified by the inverse participation ratio (IPR), plotted for an individual trajectory in Fig. 4a,

$$\mathrm{IPR}(t) = \frac{1}{\sum_{l=1}^{M} |u_l(t)|^4}, \qquad (8)$$

where $u_l$ are the expansion coefficients of the wavefunction in the basis of time-dependent HOMOs of the $M$ rubrene molecules constituting the crystal lattice. The center of charge of the hole wavefunction along the high mobility crystallographic direction $a$ (Fig. 4c), is given by

$$X_a(t) = \sum_{l=1}^{M} |u_l(t)|^2 x_a^l(t), \qquad (9)$$

where $x_a^l$ is the center of mass position of molecule $l$ along the $a$ direction.

## Data availability

Data supporting the findings of this manuscript are also available from the corresponding author upon request. Source data are provided in this paper.

## Code availability

The CustomXepr source code is available at https://github.com/OE-FET/customxepr and other software used is available upon request.

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

## Acknowledgements
We are grateful for the financial support from the European Research Council through a Synergy Grant SC2 (No. 610115) and an Advanced Grant (101020872) and the Engineering and Physical Sciences Research Council (EPSRC) through a Program Grant (EP/W017091/1). H.S. gratefully acknowledges the Royal Society for a Research Professorship (RSRP\R25\1004).

## Author contributions
R.L.C. and X.R. fabricated the devices and conducted the experiments. I.E.J. carried out the analysis on fluctuations of local fields. J.E. performed transient localization simulations and J.B. helped with their interpretation. S.S. developed the code for ESR experiments and analyses, and contributed to conceptualization and discussion of the project. E.D.G. and Z.W. helped with the fabrication of conventional rubrene FETs. H.S. supervised the project. R.L.C., X.R., I.E.J. and H.S. wrote the manuscript with input from all authors. All authors reviewed the manuscript.

## Competing interests
The authors declare no competing interests.
