## [Transparent Peer Review file · Nature Communications]

Long spin lifetimes of charge carriers in rubrene crystals due to fast transient localization motion

Corresponding Author: Professor Henning Sirringhaus

Parts of the reviewers' comments are confidential remarks to the editor and not revealed as we could not obtain permission to publish the reports from one of the peer reviewers.

Version 0:

Reviewer comments:

Reviewer #1

(Remarks to the Author)

This paper reports exceptionally long spin relaxation time at room temperature in hydrocarbon-based organic semiconductor crystals. Spin dynamics is important in understanding the electronic properties of organic semiconductors as well as in applications to spintronics. The authors utilized rubrene single crystals, which has weak spin-orbit coupling and high mobility and is advantageous in increasing spin relaxation time. The ESR signals were accumulated with a good S/N ratio by using ion-gel gating or integrating multiple crystals. The spin relaxation time was 10-100 times longer than in other reported organic semiconductors and had a unique temperature dependence. The origin of the spin relaxation is discussed in terms of transient localization, including simulations. Although I believe that the results have a high impact in the fields of organic electronics and spintronics, I recommend major revision before publishing the manuscript because there are shortcomings listed below.

The implication of this study should be described more clearly for the board readership of Nature Communications. The present manuscript is too specific to physical aspects of the FI-ESR in organic semiconductor crystals.

Experimental details should be described more clearly.

How were the ion gel films prepared?

What was the direction of the applied magnetic field with respect to the rubrene crystal?

What was the procedure of the ESR measurement at low temperature? Did they apply the voltage at room temperature and then cool down the device?

Labels are missing at the left axis of Figure 2b. It would be dimensionless parameters but not normalized.

In page 7, the authors assume that the fluctuating fields dominated by hyperfine interactions are isotropic. However, hyperfine interactions in organic materials are usually anisotropic. So, \bar{B}_x is not equal to \bar{B}_z . How about the angle dependence of the ESR linewidth in this study?

In the equation " $\bar{B}_{rms} = \bar{B}_x = \bar{B}_y = \bar{B}_z$ " at line 23 in page 7, the magnetic fields should be squared before taking average.

In page 10, what distinguishes the local scale and long-range transports in single crystals? Is it related to the representative length of electrostatic potential fluctuations by the ions?

EDLTs are not very stable in general. Figure S5b shows non-negligible hysteresis in their devices. Since ESR measurements need long measurement time, more information on the stability of the devices and the reliability of the ESR results is necessary.

The following paper is quite relevant to this work. It includes the gate voltage and temperature dependence of ESR linewidth for rubrene single-crystal field-effect transistors.

H. Matsui and T. Hasegawa: Mater. Res. Soc. Symp. Proc. 1154 (2009) B08-04.

<http://dx.doi.org/10.1557/PROC-1154-B08-04>

There are several typos.

At line 10 in page 2, "... region; it is this process that governs ..."

At line 9 in page 10, "sdriven"

Reviewer #2

(Remarks to the Author)

This work uses FI-ESR to study spin relaxation in rubrene. Rubrene has carbon and hydrogen, both light elements, and hence intrinsic spin-orbit interaction (which goes as the fourth power of the atomic number) is weak. However, I do not believe that there is no Rashba type spin orbit interaction (which accrues from internal electric fields and has nothing to do with the atomic number). To test whether Rashba interaction plays a significant role, the authors should measure the spin relaxation time under different electric fields. I don't see that the authors have nailed down the origin or mechanism of spin relaxation.

Equation (2) seems to be pure speculation. I do not see a convincing rationale here.

Page 11: "The decrease of mobility with increasing temperature is a clear indication of the band-like behavior predicted by transient localization". The decrease of mobility in band-like transport is usually due to phonon scattering.

Page 18: The diffusion coefficient of charge is not the same as the diffusion coefficient of spin in most cases (see J. Appl. Phys. 104, 014304 (2008)).

The fit between simulation and experiment in Fig. 3 is not good which calls into question the validity of the assumed Redfield mechanism.

Finally, my major concern is that I do not see any significant advance here as far as spin transport in organics is concerned. Far longer relaxation times have been reported in organic molecules or semiconductors. In fact, the statement in the abstract "In contrast to previous measurements on other molecular and polymer semiconductors, we observe remarkably long spin relaxation times—on the order of microseconds—from 15 to 290 K" is somewhat ignorant. These authors should know that much longer times [milliseconds to seconds] in similar temperature ranges have been reported in very similar systems [e.g. Nature Nanotech., 2, 216 (2007)] and also ref. [15].

Reviewer #3

(Remarks to the Author)

Long spin relaxation times of charge carriers in rubrene molecular crystals due to fast transient localization motion by R. L. Carey, X. Ren, I. E. Jacobs, J. Elsner, S. Schott, E. Goldberg, Z. Wang, J. Blumberger, and H. Sirringhaus submitted to Nature Communications

Carey et al. discuss herein on quantitative measurement of charge carrier spin relaxation in a rubrene single crystal by field-induced electron spin resonance technique. The analysis has been sophisticatedly done in particular on the relaxation dynamics, and the temperature dependence of the relaxation seems to be reasonable for those predicted in hydrocarbon-based conjugated molecules with extremely long relaxation time at RT. This work will secure the utility of the simple CH-based conjugated molecules for materials platforms in spin transport. The authors' previous excellent work published in the same Nature Communication (<https://doi.org/10.1038/s41467-023-43505-7>) employed the same technique, but reported contrasted results in the relaxation processes: the advantage of the present work but one can deduce smoothly as the reality of pure hydrocarbon molecules.

Technically one of the highlights of the present paper is an use of ion gels to inject charge carriers into single crystal of rubrene under the wide range of temperature suppressing damages of the crystal from mismatches of thermal expansion coefficients. This is smart, but as in the list of references, FI-ESR measurements of single crystal rubrenes have been already reported; technical advantages of the present work are a bit limited.

Moving onto the last section of field-effect transistor characteristics, I was much expected to the system realizing in-situ measurement of ion gel-gated FET and ESR characteristics in one identical rubrene crystals, but the measurements were done in the separatory and significantly different device configuration. I agree that, a general single crystal FET device via conventional gating is not enough to obtain reliable signals of ESR, but by gating and channel length optimization, still it might be accessible as authors states it "challenging".

Overall the work is very insightful as a piece of works in materials chemistry, but the novelty is limited in both findings and technical issues. I feel only after convincing discussions given on the following points, the work may deserve publication in Nature Communication.

1. The constant in equation 2, the precise discussions are given in supporting note 2, however physical insights on the temperature-independent factors are not enough in both the note and in the manuscript. This is the key part of the present work, and after clarifying the factors, one can go to the later discussions.
2. Temperature range of the measurements under the modulated bias applied differs significantly in particular at -0.4 V applied with the smaller charge carrier injection. It should be reexamined in the revised version.
3. The constant fraction shift dramatically in the varied bias applied, hence the different charge carrier density at the interfaces. FET measurement results may give some insights on this point, but the device configuration is substantially

different as pointed out above. Thus all the discussions are qualitative, and no clear rationale for this shift given in this manuscript.

Version 1:

Reviewer comments:

Reviewer #1

(Remarks to the Author)

The authors have fully addressed my concerns.

Reviewer #2

(Remarks to the Author)

The responses to this reviewer's comments are not entirely satisfactory.

First, these authors should realize that there is no such thing as a unique spin relaxation time. Apart from the normal T1 and T2 times, the spin relaxation time depends on how it is defined and/or measured. There is a "transport spin relaxation time" that is measured with transport effects such as in spin valves, there is "an optical spin relaxation time" that can be measured in optically active organics with polarization sensitive spectroscopy, there is a "spin phase relaxation time" which is usually measured with ESR and it even appears that the measured spin relaxation time depends on whether it is measured in bulk or in a single molecule (or few molecule clusters) [see Appl. Phys. Lett., 98, 063109 (2011)]. This is a complicated business and not so simple as the authors think it to be. As a result, there is a very wide range of values reported for the spin relaxation time in an organic spanning a few orders of magnitude. The way these authors have staked their claim is ambiguous since they have not made it clear what type of spin relaxation they are alluding to. There are multiple types.

The Rashba effect has nothing to do with material inversion symmetry unlike the Dresselhaus effect but has to do with structural inversion symmetry which is broken by an electric field. The authors recognize this. So, we need knowledge of the electric fields inside the structures. Telling the reviewer that the electric fields in EDLT and FETs are vastly different does not convince anyone. Have they measured them? One really would have to solve the Poisson equation to get the electric fields (which are spatially varying) if they do not have measurements. Also, the fact that the spin relaxation times are relatively insensitive to gate voltage is not a convincing argument. Is the range of gate voltage sufficient? How would they know? So, this response is unconvincing. The origin of spin relaxation in organics has always been controversial; some believe it is Rashba, some believe it is hyperfine interaction, etc. These authors have not even attempted to pin down the cause. If you do not even know the cause for sure, how can you make a model and then make claims? This is speculative.

Is there a theory for decrease of mobility with increasing temperature in transient localization? If there is one, why have the authors not compared theory with experiment?

Einstein relation between mobility and diffusion coefficient is valid only at equilibrium. The electric fields are quite high, and I am hard pressed to believe that these authors are anywhere close to equilibrium.

I could go on and on, but this work seems to be a mixture of actual results and speculations or hand waving. There are too many speculations, and I have mentioned only a subset of them. This work has not convinced me.

Reviewer #3

(Remarks to the Author)

The revised notations on Equation S2 in this revised manuscript are satisfactory to allow all the readers to understand and trace the story of this manuscript.

The temperature dependence reexamined in this revised version is a bit shifted from I expected, and the fit line of the signals makes no sense without fitting protocols, but I agree undetectable signals in this regime.

If the figure is not included in the manuscript, I feel the manuscript is ready for publication in this stage.

Version 2:

Reviewer comments:

Reviewer #2

(Remarks to the Author)

There is a serious contradiction between Equation (1) in the response [Poisson's equation] and the electric field plots in Fig. R4. The fields are extremely high (~ 0.1 V/nm = 100 MV/m). There is no reason to believe that equilibrium conditions hold at

such high fields. Yet Equation (1) is based on the assumption of equilibrium because the expression used for the carrier concentration assumes Boltzmann distribution which is not valid far out of equilibrium. Even if we were to ASSUME arbitrarily that equilibrium prevails, what still remains questionable is the assumption of Boltzmann statistics here instead of Fermi-Dirac statistics. How do the authors know that the organic semiconductor is non-degenerate so they can approximate Fermi-Dirac with Boltzmann? They will have to determine the position of the Fermi level with respect to the HOMO or LUMO edge.

A paper in Nature Communications is expected to be rigorous and not based on assumptions that cannot be fully substantiated. This is my major concern with this work. There is significant amount of work for which the authors must be complimented, but some of it appears to be on shaky ground and the authors would do well to perform a more thorough job. As I have mentioned earlier, I am not completely convinced by the arguments here.

Reviewer #3

(Remarks to the Author)

As the reviewer 2 says, the electric field strength of 0.1 V/nm is extremely high, making it hard to assume the equilibrium. Certainly the considerable shift in Fermi level can be presumed in the case.

But this is the case of injection of charges into the band of DNBDT, and non-degenerate nature of the band may be presumed even in the case of the high electric field applied.

The matter is in the band width of the present system, and I know it is very hard to quantify at the interface; but it might be possible to provide the width in bulk phase, and estimate the shift in EF.

Version 3:

Reviewer comments:

Reviewer #2

(Remarks to the Author)

I compliment the authors for doing the work to justify some of the assumptions. My concern was not so much the difference between Fermi-Dirac and Boltzmann, but the assumption of quasi equilibrium at high fields (both Fermi-Dirac and Boltzmann are equilibrium statistics which may not describe the situation at high fields). The correct way to handle this would be to do a transport simulation using such techniques as non-equilibrium Green's function and find the actual distribution function which may look nothing like Boltzmann or Fermi-Dirac. I understand that is too much work for these authors and therefore I will not insist on it.

This work is not entirely based on solid grounds, but has some materials that I think would be publishable. I leave that to the discretion of the editor.

Reviewer #3

(Remarks to the Author)

I have gone through the authors' reply to the comments from mine as well as the reviewer 2. The key points of discussions are in the simulation with COMSOL to gain insights into the charges accumulated at the interfaces. The parameters employed in the simulation are based on the findings in some publishes works, convincing the readers on the outcomes, I feel.

The deviation from Boltzmann from Fermi-Dirac ones is not remarkably larger as seen in the figure, suggesting the hypothesis in this work is reliable enough. Changes made in this version of the manuscript are also given on the simulation, so I feel the most of all issues rised in the previous round of revieweing are cleared.

Response to the Referees: Long spin relaxation times of charge carriers in rubrene molecular crystals due to fast transient localization motion

We thank the reviewers for their helpful and constructive comments. Below we provide a point-by-point response in blue to the comments raised and a description of the changes in the manuscript. A pdf file of the manuscript with all changes indicated in red and a related Supplementary Information file have been submitted.

Reviewer 1

This paper reports exceptionally long spin relaxation time at room temperature in hydrocarbon-based organic semiconductor crystals. Spin dynamics is important in understanding the electronic properties of organic semiconductors as well as in applications to spintronics. The authors utilized rubrene single crystals, which has weak spin-orbit coupling and high mobility and is advantageous in increasing spin relaxation time. The ESR signals were accumulated with a good S/N ratio by using ion-gel gating or integrating multiple crystals. The spin relaxation time was 10-100 times longer than in other reported organic semiconductors and had a unique temperature dependence. The origin of the spin relaxation is discussed in terms of transient localization, including simulations. Although I believe that the results have a high impact in the fields of organic electronics and spintronics, I recommend major revision before publishing the manuscript because there are shortcomings listed below.

Response: We thank the reviewer for their positive feedback and for suggesting that it might be suitable for publication in Nature Communications. We address their comments point-by-point below.

Comment 1.1

The implication of this study should be described more clearly for the board readership of Nature Communications. The present manuscript is too specific to physical aspects of the FI-ESR in organic semiconductor crystals.

Response: We believe that our study is interesting to the broader condensed matter physics community, because microsecond-long spin lifetimes at room temperature are highly unusual for mobile charge carriers in condensed matter systems. Theoretically, very long spin lifetimes are expected in material systems where both spin-orbit coupling and hyperfine interactions are weak. Graphene, one of the most widely studied spintronic materials, is an example of such a system. Unfortunately, the longest experimentally observed spin lifetimes in graphene are only on the order of 10 ns, much shorter than the theoretically predicted values (microseconds to milliseconds). The discrepancy likely arises from graphene's unique electronic properties, highlighting the challenge of achieving long spin lifetimes in semiconductors at room temperature.

Our work clearly demonstrates that despite the failure to obtain long spin lifetimes in graphene, the combination of weak spin-orbit coupling and high carrier mobility remains a feasible solution for achieving long spin lifetimes at room temperature. In materials like rubrene, weak spin-orbit coupling minimizes the impact of momentum scattering on spin relaxation, while high carrier mobility extends spin lifetimes by averaging out local hyperfine fluctuations. Further improvements in carrier mobility could potentially lead to even longer spin lifetimes. The combination of weak spin-orbit coupling and high carrier mobility could be a materials design guideline going beyond organic semiconductors.

We have revised the Introduction and Conclusions to incorporate these discussions. We hope these revisions clarify the implications of our study for the broad readership of Nature Communications.

Changes to the manuscript: We have revised the Introduction section on Page 4 to summarize the previous efforts on obtaining long spin lifetimes in materials like graphene: "Spin-orbit effects are notoriously weak in hydrocarbon-based molecules, meaning they should also have correspondingly long spin lifetimes. However, experimental observation of long spin lifetimes at room temperature is exceptionally rare. Graphene, one of the most well-known carbon materials, is theoretically predicted to exhibit spin lifetimes of microseconds to milliseconds due to its weak intrinsic spin-orbit coupling and negligible hyperfine interaction, and it is indeed one of the most extensively studied materials in spintronics.[26, 27] Yet, experimentally measured spin lifetimes in graphene are orders of magnitude smaller than those predictions, with maximum values on the order of 10 ns.[28] This gap between theory and experiment has been attributed to both extrinsic factors (e.g., substrate-induced electron-hole puddles[29]) and new spin relaxation mechanisms (e.g., resonant scattering due to magnetic impurities[30], spin-pseudospin entanglement[31]). These explanations are based on the unique properties of graphene, implying that the difficulty in achieving long spin lifetimes in graphene does not rule out the possibility of doing so in other organic semiconductors. Rubrene holds the distinction..."

We have also revised the Conclusions section to emphasize that we have discovered a mechanism for achieving long spin lifetimes at room temperature and it could potentially be a materials design guideline going beyond organic semiconductors: "We have experimentally identified a mechanism for achieving long spin lifetimes in semiconductors at room temperature, which relies on the combination of weak spin-orbit coupling and high carrier mobility. Weak spin-orbit coupling minimizes the impact of momentum scattering on spin relaxation, while high carrier mobility extends spin lifetimes by averaging out local hyperfine fluctuations. Our analysis within the transient localization framework shows that by optimising crystal quality and reducing the influence of charge carrier trapping it might be possible to achieve even longer spin relaxation times in rubrene, potentially by an order of magnitude. We believe that the combination of weak spin-orbit coupling and high carrier mobility could serve as a guideline for developing materials with long spin lifetimes, extending beyond the realm of organic semiconductors.

Our work provides new insight into the spin relaxation physics of high-mobility molecular crystals with low spin-orbit coupling and constitutes the first observation in organic semiconductors of a spin relaxation regime, in which fast, local transient localization motion of spins gives rise to effective motional narrowing, and the spin relaxation times T_1 and T_2 increase in tandem as charge motion becomes faster. In this regime charge motion occurs faster than the Larmor frequency and exceptionally long spin relaxation times are achievable, even at room temperature. We have also demonstrated that the combination of FI-ESR experiments and transient localization simulations provides a powerful approach for directly probing charge carrier motion in the unique charge transport regime that is found in such molecular crystals."

Comment 1.2

Experimental details should be described more clearly. How were the ion gel films prepared? What was the direction of the applied magnetic field with respect to the rubrene crystal? What was the procedure of the ESR measurement at low temperature? Did they apply the voltage at room temperature and then cool down the device?

Response: The ion gel solution was prepared by mixing PVDF-HFP, ionic liquid [BMP][TFSI], and acetone with a weight ratio of 1:4:20, and the ion gel films were drop-casted on glass substrates. The direction of the applied magnetic field was perpendicular to the device plane. For low temperature ESR measurements, the gate voltage was applied at room temperature and then the device was cooled down.

Changes to the manuscript: We have updated the Methods section to include these experimental details.

"...then affixed with a small drop of silver paint. The ion gel solution was prepared by mixing poly(vinylidene fluoride-co-hexafluoropropylene) (PVDF-HFP), ionic liquid 1-butyl-1-methylpyrrolidinium bis(trifluoromethylsulfonyl)imide ([BMP][TFSI]), and acetone with a weight ratio of 1:4:20, and the ion gel films were drop-cast onto glass substrates. A thin slice..."

"...and a Keithley 2612B Source-Measure Unit was used for electrical characterization. The direction of the applied magnetic field was perpendicular to the device plane. CustomXep, a Python package..."

"...The range of observed minimum values was +0.4 V to -0.6 V. For low temperature ESR measurements, the gate voltage was applied at room temperature and then the device was cooled down with the gate voltage applied."

Comment 1.3

Labels are missing at the left axis of Figure 2b. It would be dimensionless parameters but not normalized.

Response: We have updated Figure 2b to fix this issue.

Changes to the manuscript: The updated Figure 2b now has y-axis labels.

Comment 1.4

In page 7, the authors assume that the fluctuating fields dominated by hyperfine interactions are isotropic. However, hyperfine interactions in organic materials are usually anisotropic. So, $\overline{\mathcal{B}_z}$ is not equal to $\overline{\mathcal{B}_z}$. How about the angle dependence of the ESR linewidth in this study?

Response: While hyperfine interactions in organic materials may be anisotropic, our equations are derived for the motional narrowing regime. Here, charge carriers move quickly enough to average-out local hyperfine variations; because they are able to quickly sample different sites and therefore different local hyperfine distributions, the average result can be taken as equal along the three axes. This follows the work of Refs. [7], [9], [13], and [25]. Additionally, the ESR linewidth in rubrene has been reported to be almost angular-independent, as shown in Ref. [11].

We fit the data shown in Figure 3 assuming both isotropic and axially anisotropic fluctuating fields, however the anisotropic fits yielded $\overline{\mathcal{B}_{xy}} = \overline{\mathcal{B}_z}$ to within the fit uncertainty ($< 10\%$), consistent with the references above. At higher gate voltages, where spin-spin coupling becomes significant, this is no longer the case, as discussed in Supplementary Note S3.

Comment 1.5

In the equation " $\overline{\mathcal{B}_{rms}} = \overline{\mathcal{B}_x} = \overline{\mathcal{B}_y} = \overline{\mathcal{B}_z}$ " at line 23 in page 7, the magnetic fields should be squared before taking average.

Response: We have fixed this problem according to the reviewer's suggestion.

Changes to the manuscript: Now this equation is " $\overline{\mathcal{B}_{rms}^2} = \overline{\mathcal{B}_x^2} = \overline{\mathcal{B}_y^2} = \overline{\mathcal{B}_z^2}$ ".

Comment 1.6

In page 10, what distinguishes the local scale and long-range transports in single crystals? Is it related to the representative length of electrostatic potential fluctuations by the ions?

Response: Yes, it is related to the length scale of electrostatic potential fluctuations induced by the ions. In rubrene EDLTs, it has been known that conductivity (and mobility) on the device scale (long-range) is affected by the choice of ions. Therefore, although the local charge motion can be very fast, we believe that ion-induced electrostatic potential fluctuations lead to relatively low long-range charge carrier mobility.

Changes to the manuscript: We have stated in Supplementary Note S2 that the local, transient localization motion and the long-range, thermally activated transport are distinguished by the inter-ion distance (L_i).

Comment 1.7

EDLTs are not very stable in general. Figure S5b shows non-negligible hysteresis in their devices. Since ESR measurements need long measurement time, more information on the stability of the devices and the reliability of the ESR results is necessary.

Response: First, we politely disagree with the reviewer on the stability of EDLTs. In fact, EDLTs can be stable. For example, in rubrene EDLTs, multiple consecutive gate voltage sweeps give almost overlapping transfer curves with very small hysteresis (see J. Phys. Chem. C 2011, 115, 14360). In our ESR experiments, we used relatively low gate voltages (below 1.5 V) to prevent any electrochemical reactions, and there was

no stability issue (see Supplementary Note S1). The hysteresis in Figure S5b is likely due to the slow motion of ions instead of any unstable behaviour. If we further decrease the gate voltage scan rate, the hysteresis would be reduced. Second, we didn't take ESR scans while recording transfer curves. Instead, before taking ESR scans, we first set the gate voltage and hold it for several minutes to make sure that the double layers have formed and the device is at equilibrium. Consequently, the reliability of our ESR results is not affected by the hysteresis in the transfer curve.

Changes to the manuscript: We have added a short discussion in Supplementary Note S4 to clarify this point:

"...carrier concentrations greater than 10^{13} cm^{-2} . The transfer curve in Figure S5(b) shows hysteresis, which is likely due to the slow motion of ions instead of any unstable behaviour. If we further decrease the gate voltage scan rate, the hysteresis would be reduced. The reliability of our ESR results is not affected by the hysteresis in the transfer curve, because we didn't take ESR scans while recording transfer curves. Instead, before taking ESR scans, we first set the gate voltage and hold it for several minutes to make sure that the double layers have formed and the device is at equilibrium."

Comment 1.8

The following paper is quite relevant to this work. It includes the gate voltage and temperature dependence of ESR linewidth for rubrene single-crystal field-effect transistors. H. Matsui and T. Hasegawa: Mater. Res. Soc. Symp. Proc. 1154 (2009) B08-04. <http://dx.doi.org/10.1557/PROC-1154-B08-04>

Response: Thank you for pointing out the absence of this reference. We have added it as Reference [36].

Comment 1.9

There are several typos. At line 10 in page 2, "... region; it it this process that governs ..." At line 9 in page 10, "sdriven"

Response: We have fixed these typos in the revised manuscript.

Reviewer 2

This work uses FI-ESR to study spin relaxation in rubrene. Rubrene has carbon and hydrogen, both light elements, and hence intrinsic spin-orbit interaction (which goes as the fourth power of the atomic number) is weak. However, I do not believe that there is no Rashba type spin orbit interaction (which accrues from internal electric fields and has nothing to do with the atomic number). To test whether Rashba interaction plays a significant role, the authors should measure the spin relaxation time under different electric fields. I don't see that the authors have nailed down the origin or mechanism of spin relaxation.

Response: We thank the reviewer for evaluating our manuscript. We address their comments point-by-point below.

Rashba-type spin-orbit interaction usually occurs in materials with broken inversion symmetry. Rubrene crystals used in our devices are in the orthorhombic phase with inversion symmetry (space group $Cmca$). Additionally, unlike materials with strong Rashba effects (Nat. Commun. 2016, 7, 11621; Nat. Commun. 2019, 10, 4765), there is no Rashba-type band splitting feature in the rubrene band structure revealed by ARPES (Phys. Rev. Lett. 2010, 104, 156401; Appl. Phys. Lett. 2010, 96, 222106). Moreover, we can not find any experimental observation of strong bulk Rashba effect in materials consisting of only hydrogen and carbon in the literature, so there should be no bulk Rashba effect in rubrene.

In materials with inversion symmetry, the Rashba effect can still be observed if the inversion symmetry is broken by the electric field at the interface. As mentioned by the reviewer, the strength of the Rashba effect would depend on the electric field, so if the Rashba effect plays a significant role, we would observe very different spin lifetimes under different electric fields. However, this is not supported by our existing results: first, the electric field is on the order of 1 V/nm for EDLTs and 0.1 V/nm for conventional FETs, while their spin lifetimes are not significantly different (both are on the order of 1 μs); second, the spin lifetimes are nearly independent of gate voltage (electric field), as shown in Figure 6. In addition to our results, The gate voltage dependence of ESR linewidth reported in ref. [11] is also very weak. In summary, we believe the

influence of Rashba-type spin-orbit interaction should not be significant in rubrene.

Changes to the manuscript: We have included the discussion above as a new section in the Supplementary Information (Supplementary Note S9). We also mentioned this point in the main text, after showing the spin relaxation times as a function of gate voltage (Page 17):

”...in Device 1, 2, and 3, respectively. Additionally, the weak gate voltage dependence of T_2 further indicates that Rashba-type spin-orbit interaction does not play an important role, as expected from the structure of rubrene (Supplementary Note S9).”

Comment 2.1

Equation (2) seems to be pure speculation. I do not see a convincing rationale here.

Response: We acknowledge that Equation 2 is phenomenological but it has in fact a clear physical justification. To justify the functional form of motion frequency shown in Equation 2, we define 2 characteristic length scales: the inter-ion distance (L_i) and the charge localization length (L_c). L_i distinguishes the local, transient localization motion and the long-range, thermally activated transport. At low temperatures where thermal activation is not enough for a charge carrier to leave its local region, L_c simply equals L_i and becomes temperature independent (L_c should increase with decreasing temperature according to the transient localization framework, but it is already greater than L_i at room temperature). Consequently, the effective motion frequency for motional narrowing is also temperature independent, because the charge carrier experiences the same hyperfine fields at different temperatures. The hypothesis that L_c must be constrained by L_i at low temperatures is also consistent with the observation that in the FI-ESR measurements on the FET devices of Figure 4c we observe longer spin relaxation times, already at 200 K, than in the ion-gel gated devices. If the charge carrier can be thermally activated to leave its local region, then L_c becomes larger than L_i , and the effective motion frequency for motional narrowing depends on temperature (also thermally activated). Its functional form becomes the sum of a thermally activated term (contribution from thermally activated transport) and a constant term (contribution from local transient localization motion), with the former one dominating the temperature dependence at high temperature.

We acknowledge that the ion-gel gated rubrene is a complicated system, and at this stage we only have this phenomenological model which can describe the temperature dependence reasonably well. We believe that finding a better theory to quantitatively describe spin relaxation in such a complicated system is fundamentally important, and we hope to achieve this in future work.

Changes to the manuscript: We have revised Supplementary Note S2 to clarify this point:

”Equation S2 suggests that the effective motion frequency for motional narrowing always decreases with decreasing temperature. In ion-gel-gated rubrene, however, this is not valid at low temperatures. To justify the functional form of motion frequency shown in Equation 2, we define 2 characteristic length scales: the inter-ion distance (L_i) and the charge localization length (L_c). L_i distinguishes the local, transient localization motion and the long-range, thermally activated transport. At low temperatures where thermal activation is not enough for a charge carrier to leave its local region, L_c simply equals L_i and becomes temperature independent (L_c should increase with decreasing temperature according to the transient localization framework, but it is already greater than L_i at room temperature). Consequently, the effective motion frequency for motional narrowing is also temperature independent, because the charge carrier experiences the same hyperfine fields at different temperatures. The hypothesis that L_c must be constrained by L_i at low temperatures is also consistent with the observation that in the FI-ESR measurements on the FET devices of Figure 4c we observe longer spin relaxation times, already at 200 K, than in the ion-gel gated devices. If the charge carrier can be thermally activated to leave its local region, then L_c becomes larger than L_i , and the effective motion frequency for motional narrowing depends on temperature (also thermally activated). Its functional form becomes the sum of a thermally activated term (contribution from thermally activated transport) and a constant term C (contribution from local transient localization motion), with the former one dominating the temperature dependence at high temperature.”

Comment 2.2

Page 11: “The decrease of mobility with increasing temperature is a clear indication of the band-like behavior predicted by transient localization”. The decrease of mobility in band-like transport is usually due to phonon scattering.

Response: Phonon scattering is the limiting factor for band transport in wide-band inorganic semiconductors. In high mobility molecular organic semiconductors, however, their transfer integrals (bandwidths) are much smaller and the thermal fluctuation of transfer integral (i.e. dynamic disorder) can become comparable to the transfer integral itself. Therefore, the decrease of mobility with increasing temperature is not due to phonon scattering of charge carriers, but mainly due to the increased magnitude of dynamic disorder. The difference between semiclassical band transport and transient localization and why transient localization better describes charge transport in molecular semiconductors have been discussed thoroughly by Fratini and co-authors (Adv. Funct. Mater. 2016, 26, 2292).

Comment 2.3

Page 18: The diffusion coefficient of charge is not the same as the diffusion coefficient of spin in most cases (see J. Appl. Phys. 104, 014304 (2008)).

Response: We agree that charge and spin can have different diffusion coefficients, as it has been proposed by several groups. The paper mentioned by the reviewer is about spin diffusion in one particular system, 1D quantum wire with an infinite spin diffusion coefficient, which is very different from our material systems, so the conclusion of this paper might not be directly applicable to our work. We are also aware of more recent examples where charge and spin in organic semiconductors have different diffusion coefficients due to enhanced exchange coupling (Phys. Rev. Lett. 2013, 111, 016601), which might be more relevant to our material systems. However, this requires sufficiently small distances between charge carriers for exchange coupling, the onset of which is expected to be at charge density over 10^{20} cm⁻³ (Nat. Electron. 2019, 2, 98). Considering the charge density ($2.8 * 10^{19}$ cm⁻³) of the device in Page 18 (Page 20 of the revised manuscript), it is not likely that its charge and spin diffusion coefficients would be significantly different under this framework. Additionally, our current simple model already overestimate T_2 by 1-2 orders of magnitude. Using a much larger spin diffusion coefficient (as suggested in both J. Appl. Phys. 2008, 104, 014304 and Phys. Rev. Lett. 2013, 111, 016601) would lead to even greater deviation from the experimental results. As noted on Page 20, our calculations represent a crude, order-of-magnitude estimation of T_2 . It would be hard to accurately perform the calculations using different diffusion coefficients for charge and spin, which requires a more complex model with many additional factors. Given these considerations, we think it is reasonable to use the same diffusion coefficient for charge and spin in our simple model at the current stage.

Changes to the manuscript: We have added a few sentences (Page 20 of the revised manuscript) to acknowledge that charge and spin can have different diffusion coefficients and cited the according reference mentioned by the reviewer:

”...at which spins experience close encounters. We are also aware that charge and spin can exhibit different diffusion coefficients.[52, 53] In certain cases, spin diffusion coefficients can exceed charge diffusion coefficients in organic semiconductors due to enhanced exchange coupling.[53] However, it’s unclear if this applies to rubrene, and using a larger spin diffusion coefficient in our model would lead to even greater deviations from the experimental results. Despite these shortcomings...”

Comment 2.4

The fit between simulation and experiment in Fig. 3 is not good which calls into question the validity of the assumed Redfield mechanism.

Response: While we agree that the fit quality for T_1 in particular is not ideal, we stress that the full set of relaxation times shown in Figure 3 are fit together from only four free parameters—the three parameters describing the temperature dependence of the correlation time in Equation 2, plus \mathcal{B}_{rms} . Given the relative simplicity of the model, the agreement with experiment is to us a good indication that the Redfield mechanism provides a useful framework for understanding spin relaxation in this system. The variations in T_1 vs. T are not systematic which suggests the fit error is dominated by experimental uncertainty, which likely originates from the difference in the way T_1 and T_2 are measured. T_1 is obtained by fitting the evolution of the resonance line width and intensity as the microwave power is increased, and therefore is susceptible to error from fluctuations in microwave power or errors in cavity tuning. For example, if the sample position (which is difficult to precisely control when loading the sample into the microwave cavity) was shifted by 1 mm, there would be a 0.03 μ s change in T_1 due to the change of microwave power. T_2 , on the other hand, is derived only from the linewidth well below saturation, which is insensitive to fluctuation in these parameters. We generally argue, that there is a larger measurement error in T_1 due to the limited number of measurement

points in the power-saturation dependence used to extract T_1 compared to the direct determination of T_2 from the linewidth.

Changes to the manuscript: We have included a brief discussion of this in the figure caption of Fig. 3:

“The variations in T_1 vs. T in (a) are not systematic, which suggests the fit error is dominated by experimental uncertainty that likely originates from the difference in the way T_1 and T_2 are measured. T_1 is obtained by fitting the evolution of the resonance line width and intensity as the microwave power is increased, and therefore is susceptible to error from fluctuations in microwave power or errors in cavity tuning. For example, if the sample position (which is difficult to precisely control when loading the sample into the microwave cavity) was shifted by 1 mm, there would be a $0.03 \mu\text{s}$ change in T_1 due to the change of microwave power. T_2 , on the other hand, is derived only from the linewidth well below saturation, which is insensitive to fluctuation in these parameters, the T_2 data in (c) are therefore better described by the model.”

Comment 2.5

Finally, my major concern is that I do not see any significant advance here as far as spin transport in organics is concerned. Far longer relaxation times have been reported in organic molecules or semiconductors. In fact, the statement in the abstract “In contrast to previous measurements on other molecular and polymer semiconductors, we observe remarkably long spin relaxation times—on the order of microseconds—from 15 to 290 K” is somewhat ignorant. These authors should know that much longer times [milliseconds to seconds] in similar temperature ranges have been reported in very similar systems [e.g. Nature Nanotech., 2, 216 (2007)] and also ref. [15].

Response: We apologize that this sentence is a little bit ambiguous. What we intended to say is that the observation of microsecond-long spin relaxation times *at room temperature* in our experiments has not been achieved in other organic materials, and the spin lifetimes remain sufficiently long down to 15 K. We, however, respectfully disagree with the reviewer’s assertion that that much longer spin lifetimes in similar temperature ranges have been reported in very similar systems.

In the first paper mentioned by the reviewer (Nature Nanotech. 2007, 2, 216), spin relaxation times (*reported only below 100 K*, no room temperature value) are indirectly estimated using a vertical spin-valve structure. These estimations rely on the assumption (not direct measurement) of a temperature independent carrier mobility. While spin-valves have been commonly used to measure spin diffusion lengths, determining spin lifetimes requires independent measurements of carrier mobilities. Moreover, the organic spin-valve community has recognized that spin transport through organic molecules may not always occur in organic spin-valves, and some studies question whether the tunneling magnetoresistance effect plays a role in these devices (e.g., Appl. Phys. Lett. 2007, 90, 072506; Phys. Rev. B 2010, 81, 195214; Synth. Met. 2010, 160, 216; Appl. Phys. Lett. 2015, 106, 082408). According to these studies, extremely careful device fabrication is essential to reliably observe spin transport in organic spin-valves. In contrast, ESR is free of these issues and provides a more direct measurement of spin lifetimes compared to spin-valves. Considering the substantially different experimental techniques, it is not fair to compare the the reported values from spin-valves with the spin lifetimes directly measured from ESR in our study.

Ref. [15] reports FI-ESR measurements on a different molecular crystal, C10-DNBDT-NW. However, in this paper, long spin lifetimes are only observed *at low temperatures*, and the room temperature spin lifetimes are much shorter (10-100 ns) due to stronger spin-orbit interactions in sulphur-containing materials. Furthermore, the temperature dependence of spin lifetimes in this paper is different from our results. The substantial differences between ref. [15] and our results make our observation even more unusual. Indeed, rubrene stands out as the only organic semiconductor where such long spin lifetimes at room temperature have been observed.

Changes to the manuscript: We have revised the sentence in the abstract to make it clear that spin relaxation times on the order of microsecond *at room temperature* has not been reported in other organic materials:

”In contrast to previous measurements on other molecular and polymer semiconductors, we observe remarkably long spin relaxation times—on the order of microseconds—at room temperature and they remain sufficiently long down to 15 K.”

In the Introduction section (Page 3), We have also included a reference to the ESR technique, and emphasized that ESR provides a more direct measurement of spin lifetimes compared with other techniques:

”Field-induced electron spin resonance (FI-ESR) is a powerful probe of spin relaxation that has been used extensively to establish intricate links between charge and spin physics in organic FETs (OFETs). Compared with other techniques, such as spin-valves or spin-pumping measurements, which may suffer from device-related artifacts,[16, 17, 18, 19, 20, 21] ESR provides a more direct way to determine spin lifetimes by analyzing the dependence of the ESR signal on microwave power and magnetic field.[22]”

Reviewer 3

Long spin relaxation times of charge carriers in rubrene molecular crystals due to fast transient localization motion by R. L. Carey, X. Ren, I. E. Jacobs, J. Elsner, S. Schott, E. Goldberg, Z. Wang, J. Blumberger, and H. Siringhaus submitted to Nature Communications

Carey et al. discuss herein on quantitative measurement of charge carrier spin relaxation in a rubrene single crystal by field-induced electron spin resonance technique. The analysis has been sophisticatedly done in particular on the relaxation dynamics, and the temperature dependence of the relaxation seems to be reasonable for those predicted in hydrocarbon-based conjugated molecules with extremely long relaxation time at RT. This work will secure the utility of the simple CH-based conjugated molecules for materials platforms in spin transport. The authors' previous excellent work published in the same Nature Communication (<https://doi.org/10.1038/s41467-023-43505-7>) employed the same technique, but reported contrasted results in the relaxation processes: the advantage of the present work but one can deduce smoothly as the reality of pure hydrocarbon molecules.

Technically one of the highlights of the present paper is an use of ion gels to inject charge carriers into single crystal of rubrene under the wide range of temperature suppressing damages of the crystal from mismatches of thermal expansion coefficients. This is smart, but as in the list of references, FI-ESR measurements of single crystal rubrenes have been already reported; technical advantages of the present work are a bit limited.

Moving onto the last section of field-effect transistor characteristics, I was much expected to the system realizing in-situ measurement of ion-gel-gated FET and ESR characteristics in one identical rubrene crystals, but the measurements were done in the separatory and significantly different device configuration. I agree that, a general single crystal FET device via conventional gating is not enough to obtain reliable signals of ESR, but by gating and channel length optimization, still it might be accessible as authors states it "challenging".

Overall the work is very insightful as a piece of works in materials chemistry, but the novelty is limited in both findings and technical issues. I feel only after convincing discussions given on the following points, the work may deserve publication in Nature Communication.

Response: We thank the reviewer for their positive assessment of our manuscript and for suggesting that it may deserve publication in Nature Communications. We address their comments point-by-point below.

As mentioned by the reviewer, FI-ESR measurements of rubrene crystals have been reported, but we do not think this limits the technical advantages of our work. Previous publications only reported linewidths and g-factors of ESR signals in rubrene, mostly at room temperature. What makes our work distinct from others is the systematic measurement and detailed analysis of spin relaxation times T_1 and T_2 over a wide range of temperature, which provides valuable information about spin relaxation mechanisms and charge transport physics in rubrene.

We would like to express our appreciation for the reviewer's suggestion on in-situ measurements of ion-gel-gated FET and ESR characteristics in one identical rubrene crystal. In fact, this was what we were aiming for at the beginning of this project. However, the following difficulties prevented us from performing such measurements. It has been known that the rubrene/metal contact is critical for obtaining good transistor characteristics. The best rubrene transistors are fabricated using the so-called "flip-crystal" method, which creates a pristine, van der Waals rubrene/metal interface. To achieve perfect lamination on the solid substrate, the crystal needs to be thin and flexible, but usually the size of a thin crystal is not large enough. For a crystal with a sufficiently large area (like the one shown in Figure 1), it doesn't spontaneously stick to electrodes on a rigid substrate, making transistor measurements impractical. Contacts can also be made directly on top of the crystal by metal evaporation or by hand painting with Ag paint, but these methods also lead to non-ideal, contact-limited device with inferior performance.

Although we were unable to to perform in-situ measurements of ion-gel-gated FET and ESR characteristics in one identical rubrene crystal, we conducted such measurements in a device based on multiple thin crystals (Figure S5). Both the magnitude of spin lifetimes and the dependence on charge carrier density match well with the results shown in the main text. As a result, we believe our experiments on multi-crystal transistors and on one-crystal capacitors together provide a comparable level of information to what one would obtain from a one-crystal transistor device. We hope our clarification satisfies the reviewer.

Changes to the manuscript: We have added a new section in the Supplementary Information (Supplementary Note S8) to explain why we used one-crystal capacitors and multi-crystal transistors for FI-ESR measurements.

Comment 3.1

1. The constant in equation 2, the precise discussions are given in supporting note 2, however physical insights on the temperature-independent factors are not enough in both the note and in the manuscript. This is the key part of the present work, and after clarifying the factors, one can go to the later discussions.

Response: We acknowledge that Equation 2 is phenomenological but it has in fact a clear physical justification. To justify the functional form of motion frequency shown in Equation 2, we define 2 characteristic length scales: the inter-ion distance (L_i) and the charge localization length (L_c). L_i distinguishes the local, transient localization motion and the long-range, thermally activated transport. At low temperatures where thermal activation is not enough for a charge carrier to leave its local region, L_c simply equals L_i and becomes temperature independent (L_c should increase with decreasing temperature according to the transient localization framework, but it is already greater than L_i at room temperature). Consequently, the effective motion frequency for motional narrowing is also temperature independent, because the charge carrier experiences the same hyperfine fields at different temperatures. The hypothesis that L_c must be constrained by L_i at low temperatures is also consistent with the observation that in the FI-ESR measurements on the FET devices of Figure 4c we observe longer spin relaxation times, already at 200 K, than in the ion-gel gated devices. If the charge carrier can be thermally activated to leave its local region, then L_c becomes larger than L_i , and the effective motion frequency for motional narrowing depends on temperature (also thermally activated). Its functional form becomes the sum of a thermally activated term (contribution from thermally activated transport) and a constant term (contribution from local transient localization motion), with the former one dominating the temperature dependence at high temperature.

We acknowledge that the ion-gel gated rubrene is a complicated system, and at this stage we only have this phenomenological model which can describe the temperature dependence reasonably well. We believe that finding a better theory to quantitatively describe spin relaxation in such a complicated system is fundamentally important, and we hope to achieve this in future work.

Changes to the manuscript: We have revised Supplementary Note S2 to clarify this point:

”Equation S2 suggests that the effective motion frequency for motional narrowing always decreases with decreasing temperature. In ion-gel-gated rubrene, however, this is not valid at low temperatures. To justify the functional form of motion frequency shown in Equation 2, we define 2 characteristic length scales: the inter-ion distance (L_i) and the charge localization length (L_c). L_i distinguishes the local, transient localization motion and the long-range, thermally activated transport. At low temperatures where thermal activation is not enough for a charge carrier to leave its local region, L_c simply equals L_i and becomes temperature independent (L_c should increase with decreasing temperature according to the transient localization framework, but it is already greater than L_i at room temperature). Consequently, the effective motion frequency for motional narrowing is also temperature independent, because the charge carrier experiences the same hyperfine fields at different temperatures. The hypothesis that L_c must be constrained by L_i at low temperatures is also consistent with the observation that in the FI-ESR measurements on the FET devices of Figure 4c we observe longer spin relaxation times, already at 200 K, than in the ion-gel gated devices. If the charge carrier can be thermally activated to leave its local region, then L_c becomes larger than L_i , and the effective motion frequency for motional narrowing depends on temperature (also thermally activated). Its functional form becomes the sum of a thermally activated term (contribution from thermally activated transport) and a constant term C (contribution from local transient localization motion), with the former one dominating the temperature dependence at high temperature.”

Comment 3.2

2. Temperature range of the measurements under the modulated bias applied differs significantly in particular at -0.4 V applied with the smaller charge carrier injection. It should be reexamined in the revised version.

Response: In addition to the data shown in the manuscript, we also recorded several preliminary scans for other samples biased at -0.4 V. (Such scans are often taken as ‘test’ curves in order to determine the best spectrometer settings for the experimental dataset, such as integration time, number of scans, microwave powers, modulation amplitude, and modulation frequency.) In all preliminary scans, and in the real data, the signals at 50 K and below were either unobservable or so small that they could not be fit with our techniques (as shown in Figure R1). We therefore only report signals down to 80 K for the -0.4 V signal.

We attribute this lack of signal to the low applied bias voltage yielding only a small number of polarons in the accumulation layer, which, when combined with the shorter relaxation times at low temperatures, results in a weaker signal.

Changes to the manuscript: We have added a sentence in the caption of Figure 2 to explain the absence of the -0.4 V data at $T < 80$ K:

”the absence of the -0.4 V data at $T < 80$ K is due to the smaller number of polarons in the accumulation layer and the shorter relaxation times at low temperatures, resulting in signals that are unobservable or could not be fit with our techniques.”

Figure R1: The 50 K ESR signal for a rubrene crystal gated with ion-gel at -0.4 V.

Comment 3.3

3. The constant fraction shift dramatically in the varied bias applied, hence the different charge carrier density at the interfaces. FET measurement results may give some insights on this point, but the device configuration is substantially different as pointed out above. Thus all the discussions are qualitative, and no clear rationale for this shift given in this manuscript.

Response: As discussed in our response to Comment 3.1, this constant C is related to the length scale of the fast, transient localization motion. We believe it can be considered as an indicator for the density of counter-ions (or the inter-ion distance L_i). If the ion density is small (L_i is large), the local motional narrowing will be more efficient because the charge carriers are more delocalized, which should lead to larger spin lifetimes (and larger C) at low temperatures. Another potential effect of gate voltage is that charge carriers will be less strongly attracted and confined to the interface at lower gate voltages and will therefore undergo faster local motion due to the reduced influence of potential corrugation. In both cases, a smaller value of C is expected at a higher gate voltage, which matches the fitting results in Supplementary Note S2.

Changes to the manuscript: We have included this discussion in Supplementary Note S2:

”...and $C = 0.8, 1.8,$ and $0.9 \mu\text{s}$, respectively. At higher gate voltages, the constant C becomes smaller. As mentioned above, C is related to the length scale of the fast, transient localization motion. We believe it can be considered as an indicator for the density of counter-ions (or the inter-ion distance L_i). If the ion density is small (L_i is large), the local motional narrowing will be more efficient because the charge carriers are more delocalized, which should lead to larger spin lifetimes (and larger C) at low temperatures. Another potential effect of gate voltage is that charge carriers will be less strongly attracted and confined to the interface at lower gate voltages and will therefore undergo faster local motion due to the reduced influence of potential corrugation. Both factors would lead to a smaller C at a higher gate voltage.”

Second-Round Response to the Referees: Long spin relaxation times of charge carriers in rubrene molecular crystals due to fast transient localization motion

We thank the editor for handling our revised manuscript and the reviewers for their helpful and constructive comments. We have made the following major changes in the revised manuscript: (1) a discussion on different techniques for spin relaxation time measurements; (2) a discussion on the validity of the Einstein relation and whether our devices are at equilibrium; (3) calculating the electric field by solving the Poisson equation, and ruling out the potential influence of Rashba spin-orbit coupling as the origin of spin relaxation in rubrene; (4) other experiments and discussions requested by the reviewers (electrochemical measurements of the ionic liquid, theory for temperature-dependent mobility). Below we provide a point-by-point response in blue to the comments raised and a description of the changes in the manuscript. A pdf file of the manuscript with all changes indicated in red and a related Supplementary Information file have been submitted.

Additional comments from the editor and reviewer 3

Comment 0.1

Revise this sentence “in which fast local transient localization ” for better readability.

Response: We have revised this paragraph to improve readability.

Changes to the manuscript: The last paragraph of the Conclusions section has been revised as:

“Our work provides new insights into the spin-relaxation physics of high-mobility molecular crystals with low spin-orbit coupling. It represents the first observation of a spin-relaxation regime in organic semiconductors wherein the fast, local transient-localization motion of spins gives rise to effective motional narrowing. In this regime, charge motion occurs faster than the Larmor frequency, and the spin relaxation times T_1 and T_2 increase in tandem as charge motion becomes faster. Therefore, exceptionally long spin relaxation times are achievable, even at room temperature. We have also...”

Comment 0.2

It might be better to include the electrochemical measurements of the identical ionic liquid in SI to eliminate the stability issue of ESR measurement.

Response: We have performed cyclic voltammetry (CV) measurements on a metal-ionic liquid-metal structure using the same ionic liquid [BMP][TFSI] as used in our original experiments. Our CV results suggest that the ionic liquid is stable within the voltage range used in our ESR measurements. As shown in Figure R1, after 2 V, the current starts to increase rapidly with increasing voltage. For the 2.5 V scan, a smaller peak in the reverse scan can be observed at about 1.3 V, indicating some electrochemical reactions, and this peak becomes more obvious in the 3 V scan. We also did a stress experiment by holding the voltage at 1.5 V for 30 min (typical time for a single ESR scan), and found no obvious change in the CV curves before and after stress. We thus conclude that there is no stability issue if the gate voltage is below 1.5 V.

Changes to the manuscript: We have included Figure R1 as the new Figure S2 in the updated manuscript. We have also added the discussion of the stability of [BMP][TFSI] in Supplementary Note S2.

Comment 0.3

The applied gate bias is somewhat low as 1.5 V in case of EPR measurements unlike to the case of EDLC-FET measurements. Thus the validity of the authors’ model can be decided by whether EPR measurement was

Figure R1: **Cyclic voltammetry measurements of the ionic liquid [BMP][TFSI] at room temperature.** (a) CV curves measured at different voltage ranges. Inset shows the device structure. (b) CV curves taken before and after biasing the device at 1.5 V for 30 min. For CV measurements, the voltage is applied to the Au electrode, so the sign is opposite to the sign of the gate voltage for EDLTs.

done under the equilibrium or not. Reviewer 3 thus asks for experimental evidence on whether the system is in equilibrium.

Response: As shown by Blom et al. (Phys. Rev. Lett., 2011, 107, 066605), the classical Einstein relation can be valid even in disordered polymer semiconductors when the influence of deep traps is eliminated. Moreover, work done previously in our group has shown that the Einstein relation fits the ESR results well in disordered polymeric semiconductors measured under similar conditions (Schott et al., Nat. Phys., 2019, 15, 814). Rubrene has a much lower level of deep trap states compared to the disordered polymers used in our past work and, indeed, than those considered by Blom et al., meaning that our rubrene single-crystal devices should also be sufficiently at equilibrium.

We also note that during our ESR measurements, there is no voltage applied between the source/drain electrodes, meaning the device behaves like a MOS capacitor. In this case, the equations in classic semiconductor device textbooks describing MOS capacitors are applicable, which are based on equilibrium statistics and crucially include the Einstein relation. Furthermore, our FI-ESR measurements were performed only after stabilization of the sample temperature—typically an hour or so—while each power saturation scan took several hours to complete. This time scale, along with the reproducibility of our results, also suggests that our devices were at equilibrium for the data reported here.

Regarding the mobility value used for the Einstein relation, it is a common practice within the rubrene transistor community that a small drain voltage should be used for transistor measurements and the mobility should be extracted from the linear regime. Usually in this regime, the source-drain current is Ohmic (as shown in Figure R2), indicating the field-independent nature of carrier mobility/diffusivity. If the system were non-equilibrium, we would expect a much stronger dependence of mobility on the drain electric field, which would lead to non-Ohmic IV curves.

Changes to the manuscript: We have included the relevant discussion above as a new section in the Supplementary Information (Supplementary Note 10). In the main text, we mentioned the validity of the Einstein relation after showing Equation 7 (Page 19):

”...estimate D from the Einstein relation we estimate $T_2 = 33\text{-}330 \mu\text{s}$ (for the validity of the Einstein relation, see Supplementary Note S10 for a discussion).”

Figure R2: **Ohmic IV curves in rubrene transistors: (a) a cytop-gated conventional FET; (b) an ion-gated EDLT.**

Reviewer 1

The authors have fully addressed my concerns.

Response: Thank you for your positive evaluation and recognition of our revised manuscript.

Reviewer 2

Comment 2.1

The responses to this reviewer's comments are not entirely satisfactory.

First, these authors should realize that there is no such thing as a unique spin relaxation time. Apart from the normal T1 and T2 times, the spin relaxation time depends on how it is defined and/or measured. There is a "transport spin relaxation time" that is measured with transport effects such as in spin valves, there is "an optical spin relaxation time" that can be measured in optically active organics with polarization sensitive spectroscopy, there is a "spin phase relaxation time" which is usually measured with ESR and it even appears that the measured spin relaxation time depends on whether it is measured in bulk or in a single molecule (or few molecule clusters) [see Appl. Phys. Lett., 98, 063109 (2011)]. This is a complicated business and not so simple as the authors think it to be. As a result, there is a very wide range of values reported for the spin relaxation time in an organic spanning a few orders of magnitude. The way these authors have staked their claim is ambiguous since they have not made it clear what type of spin relaxation they are alluding to. There are multiple types.

Response: We agree with the referee that there are multiple ways to measure the spin relaxation time. Each technique has its own advantages and limitations.

A spin valve is a device that uses the spin of electrons to control electrical resistance. Its key advantage is that the spin diffusion length, which is critical for device applications, can be directly extracted from the length (or thickness) dependence of the spin valve signal. However, extraction of spin relaxation time requires a separate measurement of the diffusion coefficient (or mobility). The spin valve was the first reported organic spintronic device, but local spin valves, where spin injection and detection occur between the same two electrodes, have known issues. The organic spin valve community has recognized that some spin-valve signals observed in organic local spin valves do not come from spin transport through organic molecules, but instead from artifacts such as the tunneling magnetoresistance effect. For example, the signal in paper mentioned by the reviewer during the last round of review (Nature Nanotech. 2007, 2, 216) is most likely a tunneling magnetoresistance signal, as indicated by the almost temperature independent resistance between 100 K and 1.9 K. Non-local spin valve measurements can exclude most of the artifacts associated with local spin valve measurements and have provided reliable values in materials like graphene (Nano Lett. 2016, 16, 3533) and GaAs (Nat. Phys. 2007, 3, 197). But unfortunately, it has not been achieved in organic semiconductors yet. Another method to extract the spin relaxation time from a spin valve is the Hanle effect measurement,

where the spin valve signal is measured as a function of the out-of-plane magnetic field. However, so far no unambiguous Hanle effect has been observed in organic spin valves either.

Spin transport and relaxation can also be measured optically using polarization sensitive techniques like magneto-optical Kerr or Faraday rotation, where the change in polarization is related to the magnetization of the measured material. Usually materials with stronger spin-orbit coupling exhibit stronger rotation. The spin relaxation time can be extracted from the time-resolved rotation trace or from the Hanle depolarization due to the magnetic field. This method does not require the fabrication of the whole electronic device and has been widely used to measure spin relaxation times in many inorganic semiconductors such as GaAs (Science 2005, 309, 2191) and ZnO (Appl. Phys. Lett. 2008, 92, 162109). However, no optical spin relaxation measurement has been reported in organic semiconductors such as rubrene, probably due to their weak spin-orbit coupling.

Although spin relaxation time can be measured by different techniques, when the sample quality is high enough and the measurements are properly performed, the results should be comparable. For example, spin relaxation times on the order of 1-10 ns have been observed in graphene by both non-local spin valves (Nano Lett. 2016, 16, 3533) and ESR (ACS Nano 2020, 14, 7492), while for GaAs both non-local spin valves (Nat. Phys. 2007, 3, 197) and magneto-optical Kerr effect (Science 2005, 309, 2191) yield values of 10-100 ns.

We also agree that spin relaxation times in bulk and in a single molecule might be different. However, exploring this difference is beyond the scope of our manuscript, which focuses on field-induced ESR measurements in millimeter-sized molecular crystals. Moreover, the paper mentioned by the reviewer (Appl. Phys. Lett., 98, 063109 (2011)) only reports photoluminescence and infrared absorption of an organic molecule tris-(8-hydroxyquinoline aluminum) (Alq3), and is not relevant to spin relaxation time measurements.

Changes to the manuscript: To address the referee's concern we have included the discussion of different techniques as a new section in the Supplementary Information (Supplementary Note S1 in the updated manuscript) and mentioned it in the main text (Page 3):

"...by analyzing the dependence of the ESR signal on microwave power and magnetic field (see Supplementary Note S1 for a discussion of different techniques)."

The reviewer claims that "*The way these authors have staked their claim is ambiguous since they have not made it clear what type of spin relaxation they are alluding to*". We respectfully disagree. From the very beginning of the manuscript (the first sentence in the abstract), we have clearly stated that spin relaxation times in this manuscript are measured by ESR and there are no spin relaxation measurements by other techniques presented in the manuscript. Further, in the abstract and the second paragraph of the Introduction, we unambiguously define which types of spin relaxation we are discussing in this work, clearly defining the longitudinal and transverse relaxation times for readers who may not be as familiar with spin-relaxation physics as others. Then, in the second paragraph of the Results section, we explain precisely how FI-ESR can be used to measure these relaxation times. Equation 1 even cites the Redfield equations for spin relaxation in the motional narrowing regime, which are well-known relations that concern exactly the spin-relaxation dynamics we discuss in our manuscript.

Comment 2.2

The Rashba effect has nothing to do with material inversion symmetry unlike the Dresselhaus effect but has to do with structural inversion symmetry which is broken by an electric field. The authors recognize this. So, we need knowledge of the electric fields inside the structures. Telling the reviewer that the electric fields in EDLT and FETs are vastly different does not convince anyone. Have they measured them? One really would have to solve the Poisson equation to get the electric fields (which are spatially varying) if they do not have measurements. Also, the fact that the spin relaxation times are relatively insensitive to gate voltage is not a convincing argument. Is the range of gate voltage sufficient? How would they know? So, this response is unconvincing. The origin of spin relaxation in organics has always been controversial; some believe it is Rashba, some believe it is hyperfine interaction, etc. These authors have not even attempted to pin down the cause. If you do not even know the cause for sure, how can you make a model and then make claims? This is speculative.

Response: Because the electric field in the semiconductor near the interface is indeed difficult to measure directly, we have followed the referee's suggestion and solved the Poisson equation to obtain the electric field across the device using the method described in reference (J. Appl. Phys. 2011, 110, 014510).

In the organic semiconductor layer, the Poisson equation is expressed as:

$$\frac{d^2\psi}{dx^2} = \frac{q}{\varepsilon_{sc}} n_i \exp\left(\frac{q\psi}{k_B T}\right) \quad (1)$$

Here, ψ is the potential in the semiconductor, x is the position, q is the elementary charge, ε_{sc} is the permittivity of the semiconductor, n_i is the intrinsic carrier density which is defined as $n_i^2 = N_c N_v \exp\left(\frac{-E_g}{k_B T}\right)$, N_c and N_v are the effective density of states in the conduction band (LUMO) and valence band (HOMO), E_g is the band gap, k_B is the Boltzmann constant, and T is the temperature. A few assumptions are made to simplify the analysis: the device is considered unipolar with only one type of carrier with $q\psi/k_B T \gg 1$; there is no unintentional doping effect (such as mild oxygen doping) in the organic semiconductor and no extrinsic effect (such as interfacial trap states); the flat-band voltage is zero. The band diagram of the semiconductor-dielectric-gate structure is shown in Figure R3. When a negative gate voltage is applied, the bands are bent upward, leading to charge accumulation at the semiconductor/dielectric interface. Due to the very low intrinsic carrier concentration and the floating potential of a free surface at $x = 0$, the organic semiconductor first enters a volume accumulation regime, and the potential at $x = 0$ (ψ_0) is not the same as the intrinsic Fermi level. With increasing gate voltage, the semiconductor enters a surface accumulation regime and the potential bends more near the semiconductor/dielectric interface. The potential at the semiconductor/dielectric interface is defined as ψ_s . t_{sc} and t_i represents the thickness of the semiconductor layer and the dielectric layer, respectively.

Figure R3: **Band diagram of the semiconductor-dielectric-gate structure.**

With the boundary condition $d\psi/dx|_{x=0} = 0$, the electric field across the semiconductor layer can be obtained by integrating the Poisson equation:

$$\frac{d\psi}{dx} = \sqrt{\frac{2k_B T n_i}{\varepsilon_{sc}} (\exp(q\psi/k_B T) - \exp(q\psi_0/k_B T))} \quad (2)$$

The potential can be obtained by integrating one more time:

$$\frac{q(\psi - \psi_0)}{2k_B T} = -\ln\left[\cos\left(\sqrt{\frac{q^2 n_i}{2\varepsilon_{sc} k_B T}} \exp(q\psi_0/2k_B T) x\right)\right] \quad (3)$$

Therefore, ψ_s can be expressed as:

$$\frac{q(\psi_s - \psi_0)}{2k_B T} = -\ln\left[\cos\left(\sqrt{\frac{q^2 n_i}{2\varepsilon_{sc} k_B T}} \exp(q\psi_0/2k_B T) t_{sc}\right)\right] \quad (4)$$

The boundary condition at the semiconductor/dielectric interface is given by charge continuity where ε_i is the permittivity of the dielectric layer:

$$\varepsilon_i \frac{V_G - \psi_s}{t_i} = \varepsilon_{sc} \frac{d\psi}{dx}\bigg|_{x=t_{sc}} = \sqrt{2k_B T n_i \varepsilon_{sc} (\exp(q\psi_s/k_B T) - \exp(q\psi_0/k_B T))} \quad (5)$$

Both ψ_s and ψ_0 are dependent on the gate voltage V_G , and they can be numerically solved by combining Equations 4 and 5. The electric fields at the rubrene/dielectric interface ($E = d\psi/dx|_{x=t_{sc}}$) as a function of gate voltage for both a cytop-gated conventional FET and an ionic-gated EDLT are shown in Figure R4. With increasing gate voltage, ψ_0 saturates at a maximum value due to the accumulation of charge carriers at the semiconductor/dielectric interface and the resulting screening of the gate field. Therefore, the electric field is plotted versus $V_G - \psi_{0,max}$, which better represents charge accumulation at the interface. The highest electric fields for the FET and for the EDLT are on the order of 0.1 V/nm and 1 V/nm, which is quite close

to what we would expect from their capacitance values. Moreover, for the EDLT, within the voltage range used in our experiment, the electric field can be modulated over one order of magnitude. Since the Rashba coupling coefficient is proportional to the electric field, the strength of the Rashba effect should also change over one order of magnitude.

Next, we estimate what should we expect in our ion-gated rubrene EDLTs if the Rashba effect is playing a dominant role in spin relaxation. The Rashba coupling coefficient is inversely proportional to the square root of the spin relaxation time (ref: Sci. Rep. 2017, 7, 930), so a 10-fold increase in the Rashba coupling coefficient would lead to a *100-fold* reduction of the spin relaxation time. This is contrastingly different from what we observed in our devices (Figure 5 of the revised manuscript), where the spin relaxation time is relatively insensitive to gate voltage, suggesting that the influence of Rashba effect in our rubrene devices is negligible.

Additionally, the reviewer mentioned that the origin of spin relaxation in organics has always been controversial. We agree that in some sulfur-containing materials spin-orbit coupling is very important, and we have already noted this in the Introduction section (Page 4, "The high-temperature scattering process limiting relaxation times in C10-DNBDTNW is caused by the spin-orbit interaction"). However, this is clearly not the case for rubrene which only consists of carbon and hydrogen. Previously, by performing ESR measurements on deuterated rubrene, our group demonstrated that spin-orbit coupling is much weaker than hyperfine interaction in isolated rubrene molecules in solution (ref. [43]). In this work, we do not observe any signature of spin-orbit interaction in rubrene crystals either. Therefore, we strongly believe that spin-orbit coupling does not play an important role in spin relaxation in rubrene.

Figure R4: **Electric field at the rubrene/dielectric interface calculated from the Poisson equation.**

The following parameters commonly reported in the literature are used for our calculations. Parameters for rubrene are: $N_c = N_v = 1.46 \cdot 10^{21} \text{ cm}^{-3}$, $E_g = 2.2 \text{ eV}$, $T = 300 \text{ K}$, $\varepsilon_{sc} = 3\varepsilon_0$, and $t_{sc} = 1000 \text{ nm}$. For the cytop-gated conventional FET, $\varepsilon_i = 2.1\varepsilon_0$, and $t_i = 200 \text{ nm}$ (specific capacitance 9.3 nF/cm^2). For the ion-gated EDLT, $\varepsilon_i = 10\varepsilon_0$, and $t_i = 2 \text{ nm}$ (specific capacitance $4.4 \text{ } \mu\text{F/cm}^2$).

Changes to the manuscript: We have rewritten Supplementary Section S9 to include the discussion above.

Comment 2.3

Is there a theory for decrease of mobility with increasing temperature in transient localization? If there is one, why have the authors not compared theory with experiment?

Response: The transient localization theory predicts a power-law temperature dependence of mobility ($\mu \propto T^{-n}$) in perfectly clean crystals, with the exact value of n depending on the degree of disorder. For example, a recent simulation work suggests n is in the range of 1.2 to 1.7 for perfect rubrene crystals (DOI: 10.1126/sciadv.adr1758). However, in real experiments, the situation is more complicated due to the unavoidable existence of extrinsic sources of disorder. Both the magnitude of mobility and the exponent n are usually smaller than theoretical values. $n > 1$ has only been reported in the best rubrene air-gap transistors thanks to the very clean rubrene/air interface (e.g., J. Phys. Chem. C 2013, 117, 11522). For cytop-gated rubrene transistors, n is usually smaller due to the increased extrinsic disorder. For example, values from 0.30 to 0.42 have been reported in the literature (J. Appl. Phys. 2014, 115, 164511, ACS Nano 2013, 7, 10245). Our data in Figure 4b has an exponent of 0.37, which is comparable to literature values.

Changes to the manuscript: We have included the discussion of temperature dependence in Supplementary Section S6.

Comment 2.4

Einstein relation between mobility and diffusion coefficient is valid only at equilibrium. The electric fields are quite high, and I am hard pressed to believe that these authors are anywhere close to equilibrium.

I could go on and on, but this work seems to be a mixture of actual results and speculations or hand waving. There are too many speculations, and I have mentioned only a subset of them. This work has not convinced me.

Response: We acknowledge that the applicability of the Einstein relation between mobility and diffusion has been controversial in the past. However, Blom et al. (Phys. Rev. Lett., 2011, 107, 066605) demonstrated that the relationship is valid in disordered organic semiconductor systems when the influence of deep trap states is eliminated, while work done previously in our group has shown that the relationship fits the data well in disordered polymeric semiconductors measured under similar conditions (Schott et al., Nat. Phys., 2019, 15, 814). Rubrene has a much lower level of deep trap states compared to the disordered polymers used in our past work and, indeed, than those considered by Blom et al., meaning our rubrene single-crystal devices should also be sufficiently at equilibrium.

We also note that during our ESR measurements, there is no voltage applied between the source/drain electrodes, meaning the device behaves like a MOS capacitor. In this case, the equations in classic semiconductor device textbooks describing MOS capacitors or MOSFETs must be valid, which are based on equilibrium statistics and crucially include the Einstein relation. Further, our FI-ESR measurements were performed only after stabilization of the sample temperature—typically an hour or so—while each power saturation scan took several hours to complete. This time scale, along with the reproducibility of our results, also suggests that our devices were at equilibrium for the data reported here.

Regarding the mobility value used for the Einstein relation, it is a common practice within the rubrene transistor community that a small drain voltage should be used for transistor measurements and the mobility should be extracted from the linear regime. Usually in this regime, the source-drain current is Ohmic (as shown in Figure R2), indicating the field-independent nature of carrier mobility/diffusivity. If the system were non-equilibrium, we would expect a much stronger dependence of mobility on the drain electric field, which would lead to non-Ohmic IV curves.

Changes to the manuscript: We have included the relevant discussion above as a new section in the Supplementary Information (Supplementary Note S10). In the main text, we mentioned the validity of the Einstein relation after showing Equation 7 (Page 19):

”...estimate D from the Einstein relation we estimate $T_2 = 33\text{-}330 \mu\text{s}$ (for the validity of the Einstein relation, see Supplementary Note S10 for a discussion).”

Reviewer 3

The revised notations on Equation S2 in this revised manuscript are satisfactory to allow all the readers to understand and trace the story of this manuscript. The temperature dependence reexamined in this revised version is a bit shifted from I expected, and the fit line of the signals makes no sense without fitting protocols,

but I agree undetectable signals in this regime. If the figure is not included in the manuscript, I feel the manuscript is ready for publication in this stage.

Response: Thank you for the positive comment and for supporting the publication of our manuscript. We have revised the manuscript according to your suggestions.

Changes to the manuscript: We have uploaded the fitting protocols on Github (<https://github.com/OE-FET>). We have also moved the figure showing the fitting results for -0.4 V (Figure 3 in the previous version) to the Supplementary Information.

Third-Round Response to the Referees: Long spin relaxation times of charge carriers in rubrene molecular crystals due to fast transient localization motion

We thank the editor for handling our revised manuscript and the reviewers for their helpful and constructive comments. We have estimated electric fields using Fermi-Dirac statistics and calculated the Fermi level position according to both reviewers' comments. Below we provide a point-by-point response in blue to the comments raised and a description of the changes in the manuscript. A pdf file of the manuscript with all changes indicated in red and a related Supplementary Information file have been submitted.

Reviewer 2

There is a serious contradiction between Equation (1) in the response [Poisson's equation] and the electric field plots in Fig. R4. The fields are extremely high ($0.1 \text{ V/nm} = 100 \text{ MV/m}$). There is no reason to believe that equilibrium conditions hold at such high fields. Yet Equation (1) is based on the assumption of equilibrium because the expression used for the carrier concentration assumes Boltzmann distribution which is not valid far out of equilibrium. Even if we were to ASSUME arbitrarily that equilibrium prevails, what still remains questionable is the assumption of Boltzmann statistics here instead of Fermi-Dirac statistics. How do the authors know that the organic semiconductor is non-degenerate so they can approximate Fermi-Dirac with Boltzmann? They will have to determine the position of the Fermi level with respect to the HOMO or LUMO edge.

A paper in Nature Communications is expected to be rigorous and not based on assumptions that cannot be fully substantiated. This is my major concern with this work. There is significant amount of work for which the authors must be complimented, but some of it appears to be on shaky ground and the authors would do well to perform a more thorough job. As I have mentioned earlier, I am not completely convinced by the arguments here.

Response: We agree with Reviewer 2 that it is important to confirm whether Fermi-Dirac statistics can be approximated with Boltzmann statistics. To address this, in the revised manuscript, we performed finite element analysis to estimate electric fields using the **Semiconductor Module in COMSOL Multiphysics**, which allows us to select carrier statistics (Fermi-Dirac or Boltzmann). Details of the simulations are described below.

Figure R1: The gate-dielectric-semiconductor structure used for COMSOL simulations.

(1) The device is a gate-dielectric-semiconductor capacitor as shown in Figure R1. A gate voltage is applied to the Au gate, and the Poisson equation is solved for different gate voltages.

(2) The following parameters commonly reported in the literature are used for the semiconductor (rubrene) layer: $N_c = N_v = 1.46 \cdot 10^{21} \text{ cm}^{-3}$, $E_g = 2.2 \text{ eV}$, $E_C = 3.2 \text{ eV}$, $E_V = 5.4 \text{ eV}$, and $\epsilon_{sc} = 3\epsilon_0$. The p-type doping level is set to be 10^{15} cm^{-3} throughout the entire layer, as suggested by a recent paper (Nat. Commun. 2024, 15, 626). In the simulation, we set the thickness of rubrene to be $100 \text{ }\mu\text{m}$, but the results indicate that carrier accumulation primarily occurs within the first 1-2 nm from the rubrene/dielectric interface, as expected in a field-effect device.

(3) The left boundary of the rubrene layer (Au-dielectric stack) is modeled by the “thin insulator gate” boundary condition in COMSOL. The workfunction of Au is 5.1 eV. For the cytop dielectric, $\epsilon_i = 2.1\epsilon_0$, and $t_i = 200 \text{ nm}$ (specific capacitance 9.3 nF/cm^2). For the ionic liquid dielectric, $\epsilon_i = 10\epsilon_0$, and $t_i = 2 \text{ nm}$ (specific capacitance $4.4 \text{ }\mu\text{F/cm}^2$).

(4) The right boundary of the rubrene layer is modeled by the “metal contact” boundary condition in COMSOL. The voltage is set to be 0 V and the contact is ideal Ohmic.

(5) Both Fermi-Dirac statistics and Boltzmann statistics are used in the simulations.

The electric fields at the rubrene/dielectric interface as a function of gate voltage for both a cytop-gated conventional FET and an ionic-gated EDLT are shown in Figure R2. The highest electric fields for the FET and for the EDLT are around 0.1 V/nm and 0.25 V/nm , which are on the same order as what we would expect from their capacitance values. For the conventional FET, Fermi-Dirac statistics and Boltzmann statistics give nearly identical curves. For the ion-gated EDLT, the 2 curves match very well at low gate voltages but start to diverge at high gate voltages, because the carrier density becomes high enough that the semiconductor is no longer non-degenerate (in the simulation, the carrier density at -1 V is $2.75 \cdot 10^{13} \text{ cm}^{-2}$, on the order of 0.1 charge/molecule). But importantly, for the EDLT, within the carrier density range used in our experiment, the electric field can be modulated over one order of magnitude, regardless of the carrier statistics. Therefore, **replacing Boltzmann statistics with Fermi-Dirac statistics does not affect the conclusion of our previous version of manuscript**: the strength of the Rashba effect should be modulated over one order of magnitude. This is contrastingly different from what we observed in our devices (Figure 5 of the main text), where the spin relaxation time is relatively insensitive to gate voltage, suggesting that the influence of Rashba effect in our rubrene devices is negligible.

Figure R2: **Electric field at the rubrene/dielectric interface from COMSOL simulations.** Results for both Fermi-Dirac statistics and Boltzmann statistics at 300 K are displayed.

Next, we estimate the shift of the Fermi level as a function of carrier density. As mentioned by Reviewer 3, this is very difficult to quantify at the interface. We therefore followed Reviewer 3’s suggestion of using

the bandwidth in the bulk phase to estimate the Fermi level shift. First, we need to make some assumptions about the density of states (DOS) profile due to the lack of experimental data. A simulation by Troisi (J. Chem. Phys. 2011, 134, 034702) suggests that the DOS for a 2D rubrene lattice is weakly dependent on energy, so we assume a constant 2D DOS for our calculations. According to ARPES measurements (Phys. Rev. Lett. 2010, 104, 156401), the HOMO bandwidth of rubrene is 0.4 eV, and the 2D molecular density of rubrene is $2 \times 10^{14} \text{ cm}^{-2}$ ($4 \times 10^{14} \text{ cm}^{-2}$ carriers are needed to fully deplete the band), so the magnitude of DOS is $10^{15} \text{ cm}^{-2} \text{ eV}^{-1}$. While assuming a constant DOS might be somewhat simplistic as the real DOS should exhibit some energy dependence, we believe that it should provide reasonable order-of-magnitude information on the Fermi level shift.

Figure R3: **Calculated relationship between carrier density and Fermi level at 300 K.** Dashed lines represent carrier densities of 0.1 charge/molecule (blue, $2 \times 10^{13} \text{ cm}^{-2}$) and 0.01 charge/molecule (pink, $2 \times 10^{12} \text{ cm}^{-2}$) respectively. The corresponding positions of Fermi level with reference to the top of the HOMO band are shown in the right.

The calculated relationship between carrier density and Fermi level at 300 K is shown in Figure R3. we highlight 2 representative carrier densities: 0.01 charge/molecule ($2 \times 10^{12} \text{ cm}^{-2}$) and 0.1 charge/molecule ($2 \times 10^{13} \text{ cm}^{-2}$). At 0.01 charge/molecule, the Fermi level lies 65 meV above the top of the HOMO band, which is much larger than the thermal energy (26 meV). Consequently, Fermi-Dirac statistics can be reasonably approximated by Boltzmann statistics (the difference is about 5% at this carrier density), and the semiconductor can be treated as non-degenerate. At 0.1 charge/molecule, however, the Fermi level is 5 meV below the top of the HOMO band. In this case, the system is degenerate, and the Boltzmann approximation is not valid. These results suggest that our conventional FET devices (maximum carrier density $2.8 \times 10^{12} \text{ cm}^{-2}$) can be considered non-degenerate, while there is a crossover from non-degenerate to degenerate in ion-gated EDLT devices (maximum carrier density over 10^{13} cm^{-2}).

We would like to emphasize that in our manuscript **the classical Einstein relation is only used to estimate diffusivity near the “ T_2 peak”** in Figure 5, which corresponds to carrier densities of 2 to $3 \times 10^{12} \text{ cm}^{-2}$ (approximately 0.01 charge/molecule). As shown above, in this regime, the system can still be considered non-degenerate, and Fermi-Dirac statistics can be reasonably approximated by Boltzmann statistics. Therefore, the classical Einstein relation remains valid near the “ T_2 peak”. At carrier densities over 10^{13} cm^{-2} , the system becomes degenerate, and the classical Einstein relation needs to be replaced by the so-called generalized Einstein relation, but in our manuscript we **did not** apply the classical Einstein relation in this degenerate regime.

Another evidence for the non-degenerate nature of rubrene near the “ T_2 peak” is the Curie-like spin susceptibility from ESR measurements. If the system were degenerate, the spin susceptibility should be Pauli-like with no obvious temperature dependence, which is contrary to what we observed. Figure R4 shows the temperature dependence of bulk spin susceptibility (χ) for the same device presented in Figure S4. The inverse of susceptibility ($1/\chi$) is approximately linear with temperature, which is consistent with the Curie law. These measurements were taken at a gate voltage of -1.0 V, which is already slightly beyond the “ T_2 peak” (as shown in Figure 5, the “ T_2 peak” usually occurs between -0.5 V and -0.8 V). Therefore, it is

reasonable to conclude that the system is non-degenerate at the “ T_2 peak”.

Figure R4: **Curie-like spin susceptibility for a device at -1.0 V.** This is the same device shown in Figure S4. The temperature dependence of χ (left) and $1/\chi$ (right) indicates that the spin susceptibility is Curie-like and the system is non-degenerate.

Changes to the manuscript: We have revised the Poisson equation section and the Einstein relation section in the Supplementary Information (Supplementary Notes S9 and S10) to include the discussions above:

In Supplementary Note S9, “...under different electric fields. To address this, we performed finite element analysis to estimate electric fields using the **Semiconductor Module in COMSOL Multiphysics...** ...the strength of the Rashba effect should also change over one order of magnitude.”

In Supplementary Note S10, “It is also important to notice that the classical Einstein relation is only valid when the semiconductor is non-degenerate...
...the system is non-degenerate at the “ T_2 peak”.”

In the main text, we have mentioned these changes:

Page 16, “as expected from the structure of rubrene (see Supplementary Note S9 for a discussion on the Rashba effect).”

Page 19, “estimate D from the Einstein relation we estimate $T_2 = 33\text{-}330 \mu\text{s}$ (for the validity of the classical Einstein relation and the non-degenerate nature of the system, see Supplementary Note S10).”

Reviewer 3

As the reviewer 2 says, the electric field strength of 0.1 V/nm is extremely high, making it hard to assume the equilibrium. Certainly the considerable shift in Fermi level can be presumed in the case. But this is the case of injection of charges into the band of DNBDT, and non-degenerate nature of the band may be presumed even in the case of the high electric field applied. The matter is in the band width of the present system, and I know it is very hard to quantify at the interface; but it might be possible to provide the width in bulk phase, and estimate the shift in EF.

Response: We agree with Reviewer 3 that the system may be non-degenerate even in the case of high electric field and the key parameter is the Fermi level position. We have estimated the Fermi level position according to Reviewer 3’s suggestion of using the bandwidth in the bulk phase of rubrene.

According to the ARPES measurements (Phys. Rev. Lett. 2010, 104, 156401), the HOMO bandwidth of rubrene is 0.4 eV. Still, we need to make some assumptions about the density of states (DOS) profile due to the lack of experimental data. A simulation by Troisi (J. Chem. Phys. 2011, 134, 034702) suggests that the DOS for a 2D rubrene lattice is weakly dependent on energy, so we assume a constant 2D DOS for our calculations. Considering that the 2D molecular density of rubrene is $2 \times 10^{14} \text{ cm}^{-2}$ ($4 \times 10^{14} \text{ cm}^{-2}$ carriers are needed to fully empty the band), the magnitude of DOS is $10^{15} \text{ cm}^{-2} \text{ eV}^{-1}$. While assuming a constant DOS might be somewhat simplistic as the real DOS should exhibit some energy dependence, we believe that it should provide reasonable order-of-magnitude information on the Fermi level shift.

The calculated relationship between carrier density and Fermi level at 300 K is shown in Figure R3. we highlight 2 representative carrier densities: 0.01 charge/molecule ($2 \times 10^{12} \text{ cm}^{-2}$) and 0.1 charge/molecule

($2 \times 10^{13} \text{ cm}^{-2}$). At 0.01 charge/molecule, the Fermi level lies 65 meV above the top of the HOMO band, which is much larger than the thermal energy (26 meV). Consequently, Fermi-Dirac statistics can be reasonably approximated by Boltzmann statistics (the difference is about 5%), and the semiconductor can be treated as non-degenerate. At 0.1 charge/molecule, however, the Fermi level is 5 meV below the top of the HOMO band. In this case, the system is degenerate, and the Boltzmann approximation is not valid. These results suggest that our conventional FET devices (maximum carrier density $2.8 \times 10^{12} \text{ cm}^{-2}$) can be considered non-degenerate, while there is a crossover from non-degenerate to degenerate in ion-gated EDLT devices (maximum carrier density over 10^{13} cm^{-2}).

We would like to emphasize that in our manuscript **the classical Einstein relation is only used to estimate diffusivity near the “ T_2 peak”** in Figure 5, which corresponds to carrier densities of 2 to $3 \times 10^{12} \text{ cm}^{-2}$ (approximately 0.01 charge/molecule). As shown above, in this regime, the system can still be considered non-degenerate, and Fermi-Dirac statistics can be reasonably approximated by Boltzmann statistics. Therefore, the classical Einstein relation remains valid near the “ T_2 peak”. At carrier densities over 10^{13} cm^{-2} , the system becomes degenerate, and the classical Einstein relation needs to be replaced by the so-called generalized Einstein relation, but in our manuscript we **did not** apply the classical Einstein relation in this degenerate regime.

Another evidence for the non-degenerate nature of rubrene near the “ T_2 peak” is the Curie-like spin susceptibility from ESR measurements. If the system were degenerate, the spin susceptibility should be Pauli-like with no obvious temperature dependence, which is contrary to what we observed. Figure R4 shows the temperature dependence of bulk spin susceptibility (χ) for the same device presented in Figure S4. The inverse of susceptibility ($1/\chi$) is approximately linear with temperature, which is consistent with the Curie law. These measurements were taken at a gate voltage of -1.0 V, which is already slightly beyond the “ T_2 peak” (as shown in Figure 5, the “ T_2 peak” usually occurs between -0.5 V and -0.8 V). Therefore, it is reasonable to conclude that the system is non-degenerate at the “ T_2 peak”.

Changes to the manuscript: We have revised the Poisson equation section and the Einstein relation section in the Supplementary Information (Supplementary Notes S9 and S10) to include the discussions above:

In Supplementary Note S9, “...under different electric fields. **To address this, we performed finite element analysis to estimate electric fields using the Semiconductor Module in COMSOL Multiphysics... the strength of the Rashba effect should also change over one order of magnitude.**”

In Supplementary Note S10, “**It is also important to notice that the classical Einstein relation is only valid when the semiconductor is non-degenerate... the system is non-degenerate at the “ T_2 peak”.**”

In the main text, we have mentioned these changes:

Page 16, “as expected from the structure of rubrene (see Supplementary Note S9 for a discussion on the Rashba effect).”

Page 19, “estimate D from the Einstein relation we estimate $T_2 = 33\text{-}330 \mu\text{s}$ (for the validity of the classical Einstein relation and the non-degenerate nature of the system, see Supplementary Note S10).”

Fourth-Round Response to the Referees: Long spin relaxation times of charge carriers in rubrene molecular crystals due to fast transient localization motion

We thank the editor for handling our revised manuscript and the reviewers for their helpful and constructive comments. Below we provide a point-by-point response in blue to the comments raised in the last round of review. A pdf file of the updated manuscript and a related Supplementary Information file have been submitted.

Reviewer 2

I compliment the authors for doing the work to justify some of the assumptions. My concern was not so much the difference between Fermi-Dirac and Boltzmann, but the assumption of quasi equilibrium at high fields (both Fermi-Dirac and Boltzmann are equilibrium statistics which may not describe the situation at high fields). The correct way to handle this would be to do a transport simulation using such techniques as non-equilibrium Green's function and find the actual distribution function which may look nothing like Boltzmann or Fermi-Dirac. I understand that is too much work for these authors and therefore I will not insist on it.

This work is not entirely based on solid grounds, but has some materials that I think would be publishable. I leave that to the discretion of the editor.

Response: We thank Reviewer 2 for understanding the difficulties in performing a full transport simulation using techniques such as non-equilibrium Green's function and for all the constructive feedback during the revision. In the final revision, we have softened the claim of quasi-equilibrium at high electric fields by adding the following sentences at the end of Supplementary Note 10 (the Einstein relation section):

“A more rigorous way to prove that the system is at equilibrium would be to perform a transport simulation to find the actual distribution function. However, this would require techniques such as non-equilibrium Green's function and is beyond the scope of this manuscript.”

Reviewer 3

I have gone through the authors' reply to the comments from mine as well as the reviewer 2. The key points of discussions are in the simulation with COMSOL to gain insights into the charges accumulated at the interfaces. The parameters employed in the simulation are based on the findings in some publishes works, convincing the readers on the outcomes, I feel. The deviation from Boltzmann from Felmi-Dirac ones is not remarkably larger as seen in the figure, suggesting the hypothesis in this work is reliable enough. Changes made in this version of the manuscript are also given on the simulation, so I feel the most of all issues rised in the previous round of revieweing are cleared.

Response: We thank Reviewer 3 for supporting the publication of our manuscript and for all the constructive feedback during the revisions.